# LEX v1.6.0: A New Large-Eddy Simulation Model in JAX with GPU Acceleration and Automatic Differentiation

Xingyu Zhu[1], Yongquan Qu[2,3], and Xiaoming Shi[1,4]

[1]Division of Environment and Sustainability, Hong Kong University of Science and Technology, Hong Kong, China
[2]NSF Center for Learning the Earth with Artificial Intelligence and Physics (LEAP), Columbia University, New York, NY, USA.
[3]Department of Earth and Environmental Engineering, Columbia University, New York, NY, USA.
[4]Center for Ocean Research in Hong Kong and Macau, Hong Kong University of Science and Technology, Hong Kong, China

**Correspondence:** Xiaoming Shi (shixm@ust.hk)

**Abstract.** Large-eddy simulations (LES) are essential tools for studies on atmospheric turbulence and clouds and play critical roles in the development of turbulence and convection parameterizations. Current numerical weather models have approached kilometer-scale resolution as supercomputing facilities advance. However, this resolution range is in the so-called gray zone, where subgrid-scale (SGS) turbulence actively interacts with resolved motion and significantly influences the large-scale characteristics of simulated weather systems. Thus, a novel LES framework is required to enable the development of new SGS approaches for the gray zone. Here we used the Python library JAX to develop a new LES model. It is based on the generalized pseudo-incompressible equations formulated by Durran (2008). For a classic warm bubble case, the traditional Smagorinsky model fails to reproduce the correct structure evolution of the warm bubble, though it can modestly correct the rising speed in gray-zone resolution simulations. Utilizing the capability of JAX for automatic differentiation, we trained a deep learning-based SGS turbulence model for the same case. The trained deep learning SGS model, based on a simple autoencoder (AE), enables this physics-deep learning hybrid model to accurately simulate the expansion of the thermal bubble and the development of rotors surrounding the center of the bubble at a gray-zone resolution. The gray-zone simulation results are comparable to those of the benchmark LES resolution.

## 1 Introduction

Large-eddy simulation (LES) has been widely used in the atmospheric science community as a benchmark for the development of subgrid-scale (SGS) turbulence parameterizations in numerical weather prediction (NWP) and climate models (Teixeira and Cheinet, 2004; Sullivan and Patton, 2011; Verrelle et al., 2017; Wu et al., 2020; Jadhav and Chandy, 2021). LES has also helped the community to achieve a better understanding of cloud feedback, which interacts with boundary layer turbulence and contributes to climate sensitivity (Bretherton, 2015; Blossey et al., 2016; Tan et al., 2017; Shen et al., 2022). The capability of LES to resolve large, energy-containing turbulent eddies and model effects of SGS processes on these resolved scales, as well as their interactions with other processes such as clouds and radiation, makes it a unique and valuable tool in atmospheric science.

Although supercomputing platforms continue to advance, LES on a large domain for operational NWP is still not reachable. Current-generation regional NWP models and some regional climate simulations are often run at kilometer-scale resolution (Prein et al., 2017; Schär et al., 2020). Global simulations with kilometer-scale resolution have also been recently demonstrated for a four-month-long integration (Wedi et al., 2020).

The new challenge in kilometer-scale resolution resides in the gray zone for turbulence and convection. Gray zone (or terra incognita) is defined when the filter length scale has the same order as the dominant turbulence length scale (Wyngaard, 2004). In the gray zone, turbulence and convection can only be partially resolved and thus SGS motions interact actively with resolved motions in all three spatial dimensions and are not in statistical equilibrium, which contrasts with the assumptions used in the conventional planetary boundary layer (PBL) turbulence and cumulus convection schemes, such as horizontal homogeneity and quasi-equilibrium. As a result, the conventional parameterization schemes cannot be directly applied with such grid spacings (Chow et al., 2019; Honnert et al., 2020). Meanwhile, LES-type turbulence schemes cannot be applied to the gray zone either because they often assume isotropic turbulence and downscale energy transfer. In contrast, gray-zone turbulence is anisotropic and allows energy backscatter (Shi et al., 2019).

Therefore, LES is becoming an increasingly valuable tool for further advancement of SGS turbulence representation in gray-zone simulations and new LES codes which can run faster than before are needed to enable simulations covering large domains to capture the potential influence of SGS turbulence on the organization of convection and clouds (Shi and Fan, 2021).

For computationally intensive, highly parallelizable applications like atmospheric models, GPU-accelerated codes have been demonstrated to run much faster than conventional CPU-based implementations in Fortran or C (Demeshko et al., 2013; Price et al., 2014; Schalkwijk et al., 2015; van Heerwaarden et al., 2017; Sun et al., 2018, 2023). Recent years have seen many research efforts focused on GPU model development. Donahue et al. (2024) rewrote a GPU architecture for the Simple Cloud-Resolving Energy Exascale Earth System Atmosphere Model (SCREAM) with C($++$) and found an averaged $6\times$ acceleration compared to the CPU codes. Sridhar et al. (2022) developed Climate Machine (CliMA) with Julia and provided an architecture-portable framework for heterogeneous CPU/GPU computing for atmospheric modeling. An ocean dynamical core that can be operated on GPUs was also implemented with Julia in the newly developed ocean model, Oceananigans, and made significant achievements in model efficiency (Silvestri et al., 2024, 2025). For LES, Sauer and Muñoz-Esparza (2020) developed FastEddy, a CUDA C($++$) based model, and achieved a $6\times$ acceleration on one GPU over state-of-the-art LES using 64 CPUs.

Meanwhile, except for GPU acceleration, differentiability of LES codes is crucial for advancing next-generation deep learning (DL) based SGS parameterizations, but till now few GPU-based LES have mentioned differentiability. Differentiable LES exposes every step of the dynamical core as differentiable operations, enabling end-to-end gradient propagation through the simulation. This capability supports a more powerful training paradigm for DL parameterizations: the neural SGS module is optimized via differentiable roll-outs, adjusting its parameters based on how errors accumulate through the evolving flow dynamics. Recent years have seen a surge in the application of such coupled frameworks to physical parameterization problems (Kochkov et al., 2021; Qu and Shi, 2023; Watt-Meyer et al., 2024; Qu et al., 2024). Models trained in this way demonstrate superior forecast stability compared to both traditional schemes and offline-trained neural parameterizations. By integrating the physics-based core directly into the training loop, these hybrid approaches tend to yield more reliable and interpretable

weather and climate predictions than purely data-driven DL models. Commonly, hybrid models rely on high-fidelity numerical simulation data for training, but recent study also shows that they are available to include observational knowledge into the training process, further indicating the great potential of hybrid models to be applied for realistic simulations. For example, NeuralGCM (Kochkov et al., 2024) matches or outperforms state-of-the-art DL forecasting models across both short and long lead times, while also reducing computational cost relative to conventional general circulation models. It further improves the model performance and gives more accurate forecasts for precipitation by jointly using ERA-5 and satellite observational data (Yuval et al., 2024). These results underscore the promising application of differentiable LES for next-generation SGS parameterizations.

In this paper, a new fast and differentiable LES code that runs on GPUs is implemented with a newly developed Python library, JAX (Bradbury et al., 2018). Different from Fortran or C, numerical models written in Python codes are easier to be coupled with DL models for training. Existing work includes JAX-Fluids, a Python-based end-to-end differentiable CFD framework which is designed with JAX for compressible single and two-phase flows (Bezgin et al., 2023, 2025a), and enables end-to-end training of DL-based implicit LES models (Bezgin et al., 2025b). The new LES code is named LEX. LEX has the following distinct advantages: (1) it is numerically stable with its acoustic-wave-filtered governing equations and advanced integration schemes, (2) it computes quite fast by using XLA (accelerated linear algebra), a domain-specific compiler that accelerates code via many techniques and enables the compiled codes to run on TPUs or GPUs, (3) it is platform-agnostic, where the same code can be compiled and run on CPUs, GPUs, or TPUs, (4) it is auto-differentiable so that it enables DL-base parameterization to be trained with a coupled online training strategy (Rasp, 2020) in a physics-DL hybrid structure (von Rueden et al., 2020).

The structure of this paper is mainly organized as follows. Section 2 introduces the setup of LEX and the thermal testing case, and also the training flow of the hybrid ML-based SGS model. Validation results for LEX are shown in Section 3. Testing results for the ML model are presented in Section 4. In Section 5, the computational costs are compared to investigate to what extent LEX is faster than the conventional LES model and also the feasibility of the DL-based SGS model. Section 6 contains the summary and discussions.

## 2   Method and Experiment Design

### 2.1   LEX

#### 2.1.1   Governing Equations

To develop LEX, the acoustic-wave-filtered equations for compressible stratified flow developed by Durran (2008) are adopted as the governing equations, where a pseudo-density $\rho^*$ is defined to eliminate sound waves and enforce the mass conservation equation:

$$\frac{1}{\rho^*}\frac{D\rho^*}{Dt} + \nabla \cdot \mathbf{u} = 0, \tag{1}$$

and the pseudo-density is defined as:

$$\rho^* = \frac{\tilde{\rho}(x,y,z,t)\tilde{\theta}_\rho(x,y,z,t)}{\theta_\rho}, \tag{2}$$

where $\tilde{}$ denotes a spatially varying reference state. The potential temperature is used in the definition of (Durran, 2008) for the dry air. Here the density potential temperature $\theta_\rho$ is used as a replacement to include the effect of water variables for a moist situation. It is defined and approximated as:

$$\theta_\rho = \theta\left(\frac{1+q_v/\varepsilon}{1+q_v+q_l+q_i}\right) \approx \theta\left[1+\left(\frac{1}{\varepsilon}-1\right)q_v - q_l - q_i\right], \tag{3}$$

where $\varepsilon = R_d/R_v$, $R_d$ and $R_v$ are gas constants for dry air and water vapor, respectively. $q_v$, $q_l$, $q_i$ are mixing ratios of water vapor, liquid water, and cloud ice. In the reference state, $q_l$ and $q_i$ can be assumed to be zero, thus $\tilde{\theta}_\rho$ is the reference state virtual potential temperature.

With this definition, and further with some approximation, the mass (pseudo-density) conservation equation becomes:

$$\frac{\partial \tilde{\rho}\tilde{\theta}_\rho}{\partial t} + \nabla \cdot (\tilde{\rho}\tilde{\theta}_\rho \mathbf{u}) = \frac{\tilde{\rho}H_m}{c_p\tilde{\pi}}, \tag{4}$$

where $H_m$ is the heating rate per unit mass, $\mathbf{u}$ is velocity, $c_p$ is the specific heat of air at constant pressure, and $\pi$ is the Exner function. Perturbations with respect to the reference state are defined such that $\theta' = \theta - \tilde{\theta}$ and $\pi' = \pi - \tilde{\pi}$. Durran (2008) further separated $\tilde{\pi}$ into a large horizontally uniform component $\tilde{\pi}_v(z,t)$ and a remainder $\tilde{\pi}_h(x,y,z,t)$ for computational accuracy and notational convenience. Then the momentum and thermodynamics equations are the following,

$$\frac{D\mathbf{u}_h}{Dt} + f\mathbf{k} \times \mathbf{u}_h + c_p\theta_\rho \nabla_h(\tilde{\pi}_h + \pi') = 0 \tag{5}$$

$$\frac{Dw}{Dt} + c_p\theta_\rho \frac{\partial \pi'}{\partial z} = B \tag{6}$$

$$\frac{D\theta}{Dt} = \frac{H_m}{c_p\tilde{\pi}}, \tag{7}$$

where $\mathbf{u}_h$ is the horizontal velocity vector, $w$ is the vertical velocity, $\nabla_h$ is the horizontal gradient operator, and $f$ is the Coriolis parameter. $B$ is the linearized buoyancy,

$$B = g\left[\frac{\theta'}{\tilde{\theta}} + \left(\frac{1}{\epsilon}-1\right)(q_v - \tilde{q}_v) - q_l - q_i\right], \tag{8}$$

in which, $\tilde{q}_v$ is the reference state mixing ratio of water vapor, and $g$ is the gravitational acceleration. The reference state satisfies the equation of state and the hydrostatic balance equation:

$$\tilde{\pi} = \left(\frac{R}{p_s}\tilde{\rho}\tilde{\theta}_\rho\right)^{R/c_v} \tag{9}$$

$$c_p \tilde{\theta}_\rho \frac{\partial \tilde{\pi}}{\partial z} = -g, \tag{10}$$

where $R$ is the gas constant for dry air, $c_v$ is the specific heat of air at constant volume, and $p_s$ is the pressure at the referenced level.

The last unknown variable needed for integration is the pressure perturbation $\pi'$, which needs to be solved diagnostically to enforce Equation (4). The diagnostic relationship is obtained by multiplying the momentum equation by $\tilde{\rho}\tilde{\theta}_\rho$, taking the divergence of the result and subtracting $\partial/\partial t$ of Equation (4). The resulting diagnostic equation is provided by Durran (2008) as his Equation (5.2):

$$
\begin{aligned}
c_p \nabla \cdot (\tilde{\rho}\tilde{\theta}_\rho \theta_\rho \nabla \pi') = & -\nabla \cdot (\tilde{\rho}\tilde{\theta}_\rho \mathbf{u} \cdot \nabla)\mathbf{u} - f\nabla_h \cdot (\mathbf{k} \times \tilde{\rho}\tilde{\theta}_\rho \mathbf{u}_h) + \frac{\partial \tilde{\rho}\tilde{\theta}_\rho B}{\partial z} \\
& - c_p \nabla_h \cdot (\tilde{\rho}\tilde{\theta}_\rho \theta_\rho \nabla_h \tilde{\pi}_h) - \frac{\partial}{\partial t}\left(\frac{\tilde{\rho}H_m}{c_p \tilde{\pi}}\right) + \nabla \cdot \left(\frac{\partial \tilde{\rho}\tilde{\theta}_\rho}{\partial t}\mathbf{u}\right) + \frac{\partial^2 \tilde{\rho}\tilde{\theta}_\rho}{\partial t^2}.
\end{aligned}
\tag{11}
$$

Assuming the tendency in the reference state is small, the last few terms involving time derivative can be ignored in the equation above, then the diagnostic relation for $\pi'$ is:

$$
\begin{aligned}
c_p \nabla \cdot (\tilde{\rho}\tilde{\theta}_\rho \theta_\rho \nabla \pi') = & -\nabla \cdot (\tilde{\rho}\tilde{\theta}_\rho \mathbf{u} \cdot \nabla)\mathbf{u} - f\nabla_h \cdot (\mathbf{k} \times \tilde{\rho}\tilde{\theta}_\rho \mathbf{u}_h) + \frac{\partial \tilde{\rho}\tilde{\theta}_\rho B}{\partial z} \\
& - c_p \nabla_h \cdot (\tilde{\rho}\tilde{\theta}_\rho \theta_\rho \nabla_h \tilde{\pi}_h) = \mathcal{R}.
\end{aligned}
\tag{12}
$$

The model has no microphysics scheme yet, so water vapor is included just like a tracer, though it affects buoyancy.

### 2.1.2 Numerical Techniques

For time integration, the four-stage third-order Strong-stability-preserving Runge-Kutta (SSPRK3) scheme (Durran, 2010) is used to ensure better numerical stability. To keep numerical robustness and stability, especially for long-time integration of turbulent flows in the atmospheric boundary layer, which can develop sharp gradients and discontinuities, the weighted essentially non-oscillatory (WENO) schemes (Jiang and Shu, 1996; Shu, 1998) are employed to solve the advection tendencies for the momentum equations, with a fifth-order scheme for the horizontal direction and a third-order scheme for the vertical direction. The WENO scheme provides a proven and computationally efficient mechanism to eliminate spurious numerical oscillations. Three layers of ghost points are used in each side of $x$ and $y$ directions to employ the fifth-order WENO scheme for the horizontal fluxes. The discretization adopts the staggered Arakawa C-grid. The pressure equation (12) is solved with the biconjugate gradient stabilized method (BiCGSTAB).

### 2.1.3 Testing Simulation Configurations

The three-dimensional numerical simulation of a rising thermal (Wicker and Skamarock, 1998; Bryan and Fritsch, 2002) is applied to validate the accuracy of LEX. The employed grid spacing is 100 m in both $x$, $y$, and $z$ directions. The entire domain is

24 km by 24 km horizontally and 12 km vertically. The initial reference state has a constant potential temperature of 300 K, and features motionless air, hydrostatic equilibrium, and lapse rates corresponding to neutral stability. Periodic boundary conditions are applied to the four sides and rigid, free-slip wall boundary conditions are specified at the top and bottom of the domain. Water vapor is included for moist cases with a constant relative humidity of 10% everywhere in the initial condition. The

145 thermal is set at the central part of the domain at the bottom, with the initial potential temperature perturbation being:

$$
\theta' = \begin{cases} \theta_c \cos^2(\frac{\pi L_b}{2}) & \text{if } L_b \leq 1, \\ 0 & \text{otherwise,} \end{cases}
$$

where $\theta_c$ is used to adjust the maximum value of the potential temperature perturbation at the centre of the thermal to simulate different thermal rising speeds, and $L_b$ is the radial normalized distance between any point in the domain and the centre of the thermal, which is defined as:

$$
L_b = \sqrt{(\frac{x - x_c}{x_r})^2 + (\frac{y - y_c}{y_r})^2 + (\frac{z - z_c}{z_r})^2},
\tag{13}
$$

where $\boldsymbol{X}_c$ is the coordinates of the thermal centre, with $x_c = y_c = 12\,\text{km}$, $z_c = 2\,\text{km}$, and $\boldsymbol{X}_r$ is the initial radius of the thermal, with $x_r = y_r = z_r = 2\,\text{km}$.

The initial potential temperature perturbation will induce an upward buoyancy force and initiate the vertical acceleration of the bubble in the very beginning. The buoyancy will then cause the thermal to rise and evolve. During rising, the upper part

of the thermal will elongate. Two rotors will also be developed on each side of the bubble in this process. The structure of the thermal maintains strictly symmetric as it evolves.

## 2.2 Deep Learning SGS Model

### 2.2.1 SGS Correction

Written with JAX, DL models can be coupled with the LES model for training. This paper tested the hybrid model's capability

to use a DL-based SGS parameterization.

The SGS process refers to the unresolved part of the numerical simulations due to the relatively coarse grid size. Taking the potential temperature equation (7) for example, it can be written as the following on a numerical grid:

$$
\frac{D\bar{\theta}}{Dt} = \frac{H_m}{c_p \tilde{\pi}} + \tau,
\tag{14}
$$

where $\bar{\theta}$ is the LES grid filtered potential temperature and $\tau$ is the SGS tendency.

To improve the stability of the numerical integration, in this paper, the DL model is used to represent an SGS correction term instead of the tendency term (Um et al., 2021; Kochkov et al., 2021; Qu and Shi, 2023), which is defined as:

$$
T_\theta = \int\limits_{t_0}^{t_0 + \Delta t} \tau \, dt,
\tag{15}
$$

where $T$ is the SGS correction term and the integration is for one time step of the dynamical core. $T$ can be obtained from the DL model, which is:

$$T_\theta = \mathcal{M}(\hat{\bar{\theta}}'_{t_0+\Delta t}, \hat{\bar{q}}'_{v,t_0+\Delta t}, \hat{\bar{u}}_{t_0+\Delta t}, \hat{\bar{v}}_{t_0+\Delta t}, \hat{\bar{w}}_{t_0+\Delta t}, \hat{\bar{\pi}}'_{t_0+\Delta t}). \tag{16}$$

$\mathcal{M}$ is the DL model, $t_0 + \Delta t$ denotes that those are the variables after one time step integration of the dynamical core, and $\hat{}$ denotes that those have not been corrected by the DL model. Thus for each DL model correcting step, the forecast status of the potential temperature ($\theta$) is updated as:

$$\bar{\theta}_{t_0+\Delta t} = \hat{\bar{\theta}}_{t_0+\Delta t} + T_\theta, \tag{17}$$

and similarly, such SGS correction terms are applied to the mixing ratio of water vapor ($q_v$), the horizontal and vertical velocity ($\mathbf{u} = u, v, w$):

$$\bar{\mathbf{u}}_{t_0+\Delta t} = \hat{\bar{\mathbf{U}}}_{t_0+\Delta t} + T_{\mathbf{U}}, \tag{18}$$

$$\bar{q}_{v,t_0+\Delta t} = \hat{\bar{q}}_{v,t_0+\Delta t} + T_{q_v}. \tag{19}$$

### 2.2.2 Data, Model Structure and Training Configurations

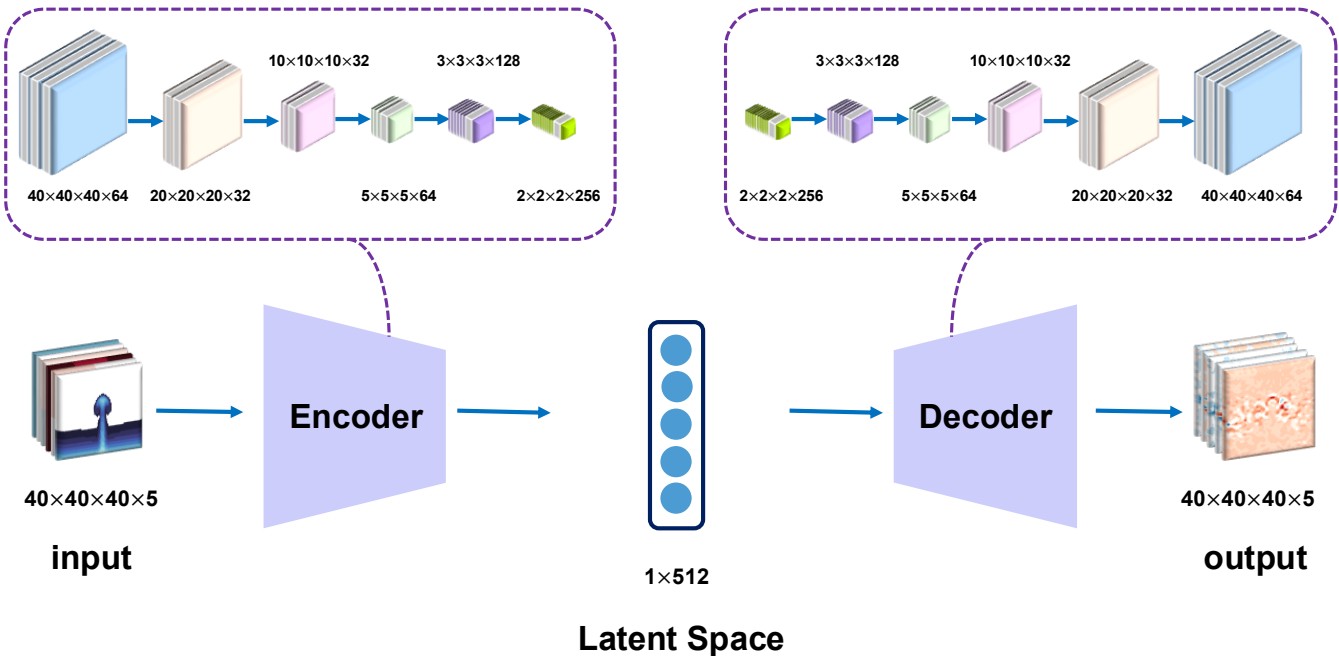

**Figure 1.** Model Architecture for the three-dimensional autoencoder neural network, where $a \times b \times c \times d$ means width$\times$length$\times$height$\times$channel. The inputs include the density potential temperature perturbation ($\theta'$), pressure perturbation ($\pi'$), mixing ratio of water vapor perturbation ($q_v'$), horizontal and vertical velocity ($u, v, w$), and the outputs are SGS corrections for the density potential temperature ($\theta$), mixing ratio of water vapor ($q_v$), horizontal and vertical velocity ($u, v, w$).

The training dataset is based on high-resolution 'truth' simulations (HighRes), which have a grid spacing of 100 m in both horizontal and vertical directions. Six distinct warm bubble cases are included in the dataset. Each is initialized with different potential temperature perturbations prescribed with $\theta_c$ at 0.5, 1.0, 1.5, 2.0, 2.5, and 3.0 K. The dataset only contains data before the warm bubble's complete interaction with the upper boundary. Accordingly, simulation durations are set to 30, 25, and 20 minutes for Case pairs 0.5–1.0 K, 1.5–2.0 K, and 2.5–3.0 K, respectively. A time step of 5 s is used for all simulations. Then by spatially coarse-graining the HighRes data, the training dataset is generated, with a grid spacing of 600 m in the horizontal and 300 m in the vertical. 90% of the generated dataset is used for training, and 10% is used for testing. Temporal coarse-graining is employed in the training and testing processes, with a 15-second time step for the numerical simulation.

To validate the trained model, two additional cases with $\theta_c = 2.6$ K and 5.0 K are chosen to generate initial conditions for validation simulations. These two initial conditions are chosen to be within the training dataset potential temperature perturbation range as well as outside that range to evaluate the capability and generalizability of the trained DL SGS model.

A three-dimensional autoencoder (3-D AE) is designed as the structure of the DL SGS, shown in Figure 1. The architecture employs 3D convolutional filters with a kernel size of $3 \times 3 \times 3$ and 'same' padding throughout its encoder and decoder hidden layers, utilizing the GELU activation function (Hendrycks and Gimpel, 2023). The model contains approximately 7.09 million trainable parameters.

A moist warm bubble case is trained in this paper. The density potential temperature perturbation ($\theta'$), pressure perturbation($\pi'$), mixing ratio of water vapor perturbation($q_v'$), horizontal and vertical velocity ($u, v, w$) are used as inputs for the AE model, with ghost points reserved to preserve the physical information at the boundaries. The outputs are SGS corrections for the density potential temperature ($\theta_\rho$), mixing ratio of water vapor ($q_v$), horizontal and vertical velocity ($u, v, w$). It should be noticed here that the outputs with ghost points are then stripped of their boundary extensions and repadded with new ghost points to maintain numerical stability. Moreover, all the physical quantities are min-max normalized, with min-max values of each height level throughout the training dataset, to a unified range of $[0, 1]$ before being input to the AE model to avoid unit-induced disparities in data distribution. The height-dependent min-max normalization for model inputs can be written as:

$$\tilde{\Phi} = \frac{\Phi - [\min(\Phi)]}{[\max(\Phi)] - [\min(\Phi)]}, \tag{20}$$

where $\Phi$ is the tensor of the general physical state consisting of density potential temperature perturbation ($\theta'$), mixing ratio of water vapor perturbation ($q_v'$), horizontal and vertical velocity ($u, v, w$). $\tilde{\Phi}$ means the normalized physical state. The max and min functions find the extreme at each grid point in the entire training dataset, and the square brackets indicate further taking the min and max values of each height level. The motivation of the height-dependent min-max normalization is to ensure the trained model has optimized performance at all levels, as the atmosphere is a stratified fluid and different height levels tend to have different amplitudes of variability, especially for thermodynamic variables.

The outputs of the DL model are multiplied by $\max - \min$ values of each grid point throughout the training dataset before they are added back to the direct integration results, which is:

$$\tilde{T}_{\Phi_{t_k}} = T_{\Phi_{t_k}} \times (\max(\Phi) - \min(\Phi)), \tag{21}$$

where $T_{\Phi_{t_k}}$ is the correction term of each variable, $\tilde{T}_{\Phi_{t_k}}$ is the scaled correction term, and the max and min functions are the extremes at each grid point in the entire training dataset. This is designed to make all the DL model's outputs suitable with the order of the direct integration results for each variable, and can also help avoid adding unnecessary corrections to points that have no variations in the training dataset.

The overall training flow can be summarized as follows. At the beginning of each training step, a numerical integration step is performed for the dynamical core from given initial states, which are coarse-grained from the high-resolution benchmark simulations. Then the integration results are used as inputs to the DL model to yield the SGS correction terms, which are applied to the direct integration results to get the new physical states. Such new physics states serve as the initial states for the next numerical integration step. This loop is iterated for $N$ look-ahead steps.

At each step, the $L_2$ loss, Laplacian loss, as well as an extra loss term which is used to penalize unreasonable model outputs, are employed and accumulated to be the total loss of the current training step, with which we use the Adam (Kingma and Ba, 2017) optimizer to adjust the DL model parameters. In this study $N = 12$. To mitigate the influence of the potential rounding errors, double-precision (float64) is employed throughout the training process.

The $L_2$ loss is written as:

$$\mathcal{L}_{l_2}^k(\Phi_{t_0}, \Phi_{t_k}) = \frac{1}{2}\|M^k \Phi_{t_0} - \Phi_{t_k}\|_2^2, \tag{22}$$

where $M$ represents the hybrid model (dynamical core and the SGS model), $\Phi_{t_0}$ is the initial state of $\Phi$, and $\Phi_{t_k}$ is the truth state of $\Phi$ at the $k^{\text{th}}$ look-ahead step during training.

When training, it is found that the DL model can not distinguish the physical meaning of the input variables and will generate unreasonable outputs in the very beginning, such as negative values for water vapor, and meaningless background noise, which are against the laws of physics and will strongly impact the integrated results of the numerical simulation in the next time step during the training loop.

Thus, firstly, to penalize such unreasonable negative values that may be caused by the DL model, an extra loss term is added, which is:

$$\mathcal{L}_{penal-neg}^k(q_{v,t_k}) = \begin{cases} -q_{v,t_k}, & \text{if } q_{v,t_k} < 0, \\ \\ 0, & \text{if } q_{v,t_k} \geq 0. \end{cases} \tag{23}$$

To mitigate background noise, a Laplacian loss as well as scaling parameters are further employed. Laplacian loss, which utilizes a Laplacian operator or Laplacian pyramid, is known to effectively improve image qualities by enhancing details and reducing noise (Li et al., 2017; Didwania et al., 2025). In this case, the Laplacian of the density potential temperature and mixing ratio of water vapor perturbations are added to loss terms, which can be written as:

$$\mathcal{L}_{Lap}^k(\Phi_{t_0}, \Phi_{t_k}) = \|\nabla^2 M^k \Phi_{t_0} - \nabla^2 \Phi_{t_k}\|_2^2 \tag{24}$$

A scaling parameter is also applied to each loss term, which is:

$$f_{scale}(\Phi) = \frac{\max(\Phi) - \min(\Phi)}{([\max(\Phi)] - [\min(\Phi)])^3}.$$

(25)

The design of this custom scaling factor is intended to introduce spatially varying normalization. Firstly, the loss terms are also height-dependently normalized, so the squared error should be divided by $([\max] - \min)^2$. Secondly, as horizontal variability is non-uniform for all variables, and near-zero perturbations occur in regions far from the thermal bubble, it does not make sense to require the deep learning SGS model to emphasize these regions — an aspect we have accounted for by scaling the SGS correction terms derived from the DL model. This scaling approach is also meaningful for real atmospheric modeling. For small simulation domains, the horizontal heterogeneity may be caused by complex terrain or land-sea contrast.

Thus to summarize, the loss function employed in this paper is written as:

$$\mathcal{L}_{tot} = \frac{1}{N} \sum_{k=1}^{N} (f_{scale}(\Phi) \cdot \mathcal{L}_{l_2}^{k}(\Phi_{t_0}, \Phi_{t_k}) + f_{scale}(q_v) \cdot \mathcal{L}_{penal-neg}^{k}(q_{v,t_k}) + f_{scale}(\nabla^2 \Phi) \cdot \mathcal{L}_{Lap}^{k}(\Phi_{t_0}, \Phi_{t_k})),$$

(26)

where $N$ is the number of the look-ahead steps, and $\Phi_{t_k}$ is $\hat{\bar{\Phi}}_{t_k} + \tilde{T}_{\Phi_{t_k}}$. It should be mentioned here that the training process is quite sensitive to the initialization of the DL model parameters. Thus it begins with three look-ahead steps as a pre-training strategy, and gradually adds up to six and twelve look-ahead steps to keep the model's numerical stability.

In this paper, the AE model is trained for 88 epochs, with the training process being manually fine-tuned based on the observed loss trajectory. The learning rate is decayed from an initial value of 1e-3 to a final value of 1e-6, which can be seen in Figure S1.

## 3   LEX Validation

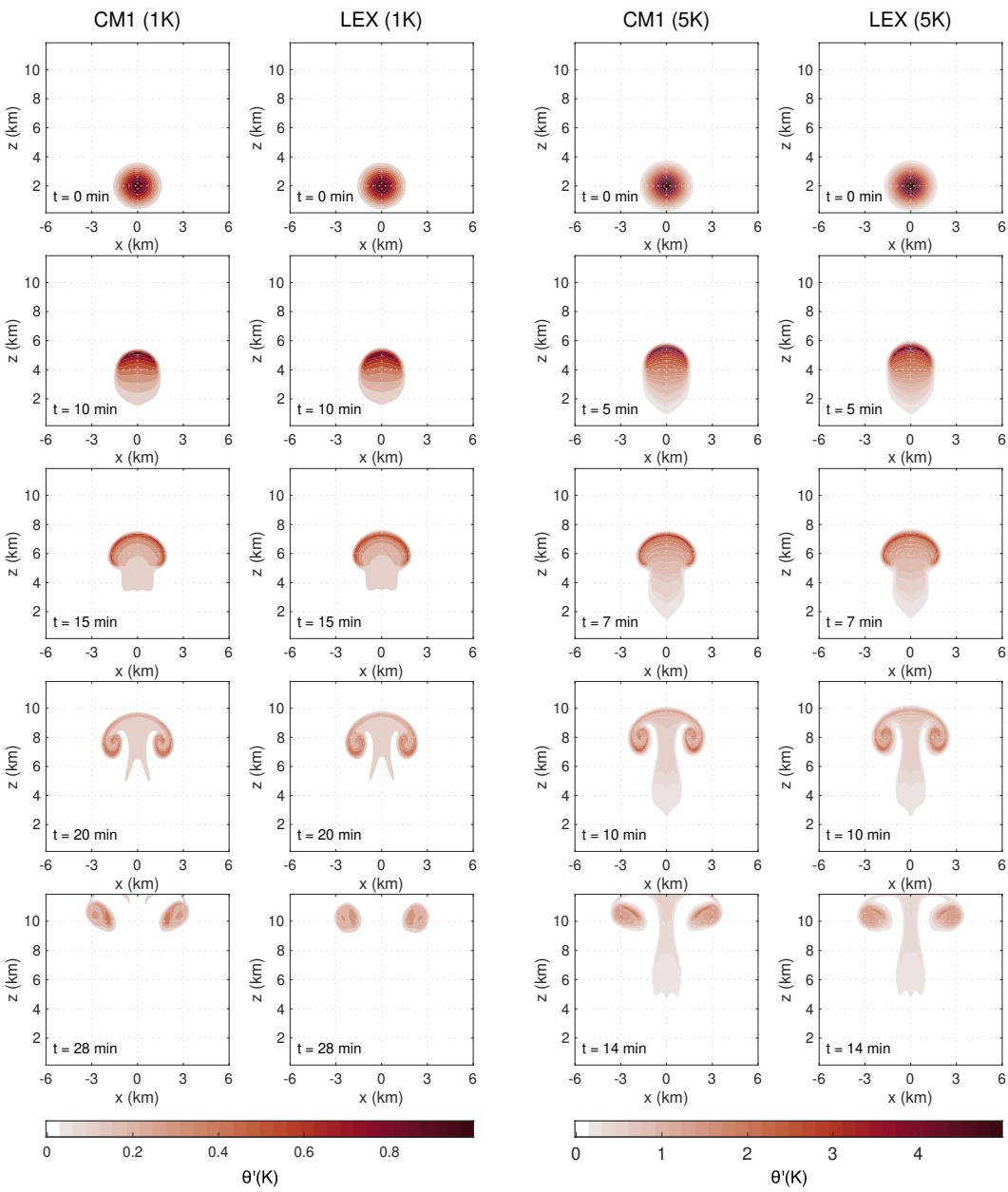

**Figure 2.** Snapshots of simulated potential temperature perturbations ($\theta'$) in CM1 and LEX under different initial central perturbations: 1 K at t = 0, 10, 15, 20, and 28 min (the first and second columns) and 5 K at t = 0, 5, 7, 10, and 14 min (the third and fourth columns).

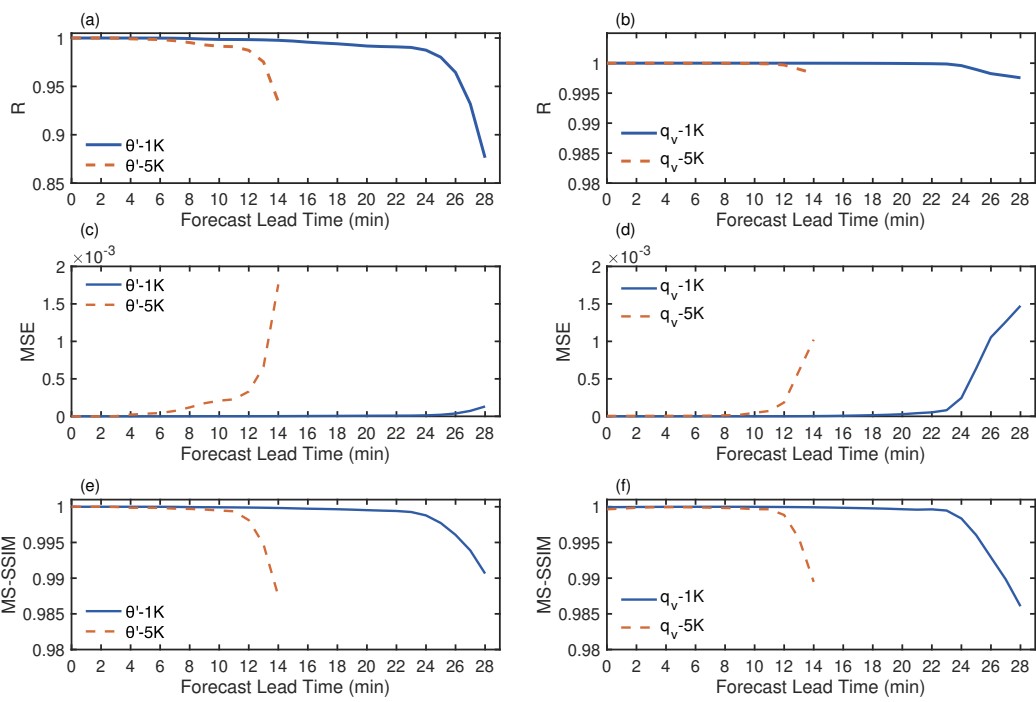

**Figure 3.** The correlation coefficient (R), mean squared error (MSE), and multi-scale structural similarity (MS-SSIM) for CM1 and LEX simulation results of density potential temperature (a, c, e), and water vapor mixing ratio (b, d, f).

The accuracy of LEX is validated against high-fidelity simulation results obtained from the fully compressible Cloud Model 1 (CM1) (Bryan and Fritsch, 2002). Two moist cases with different initial central potential temperature perturbations are tested. The first and second columns of Figure 2 show the snapshots of the simulated potential temperature perturbations in CM1 and LEX with $\theta_c = 1K$ at $t = 0, 10, 15$ and 20 min. The third and fourth columns of Figure 2 show the simulated results with $\theta_c = 5K$ at $t = 0, 5, 7, 10$ and 14 min. Comparing the two pairs in Figure 2, it is evident that the simulation results of LEX demonstrate excellent agreement with those of CM1, regardless of the initial potential temperature perturbations, indicating the reliability and accuracy of the LEX's code in JAX. Also, the results shown in Figure 3 demonstrate excellent performance metrics of LEX. The correlation coefficient (R) and the multi-scale structural similarity (MS-SSIM) maintain high values, and the mean squared error (MSE) maintains low levels throughout the simulation period for two tested cases. Figure S2 and Figure S3 further confirm the robustness of LEX by presenting the simulated results for the mixing ratio of water vapor and pressure perturbations with $\theta_c = 1K$ and $\theta_c = 5K$. However, because LEX calculates pressure based on the pseudo-compressible approximation, subtle differences appear after the thermal reaches the upper boundary of the domain in pressure simulations.

Furthermore, the figure shows that different initial potential temperature perturbations make the thermal rise at different speeds. A higher temperature brings the thermal a faster rising speed. The bubble with $\theta_c = 5\,\mathrm{K}$ takes around half of the simulated time to rise to a similar height as the bubble with $\theta_c = 1\,\mathrm{K}$. The acceleration of the rising speed due to a warmer initial potential temperature perturbation makes an evident difference for the simulated physical states of the numerical model in each integration time step. Therefore, though the evolution patterns are quite similar in the five training cases, they still provide rich and complex variation in the training dataset, sufficient for the training and testing of the DL SGS model below. The additional validation case initiated with $\theta_c = 5\,\mathrm{K}$ is far outside of the training dataset range, further proving the generalizability of the trained DL model.

## 4 Preliminary Testing for Deep Learning-based Parameterizations

### 4.1 Conventional SGS Model

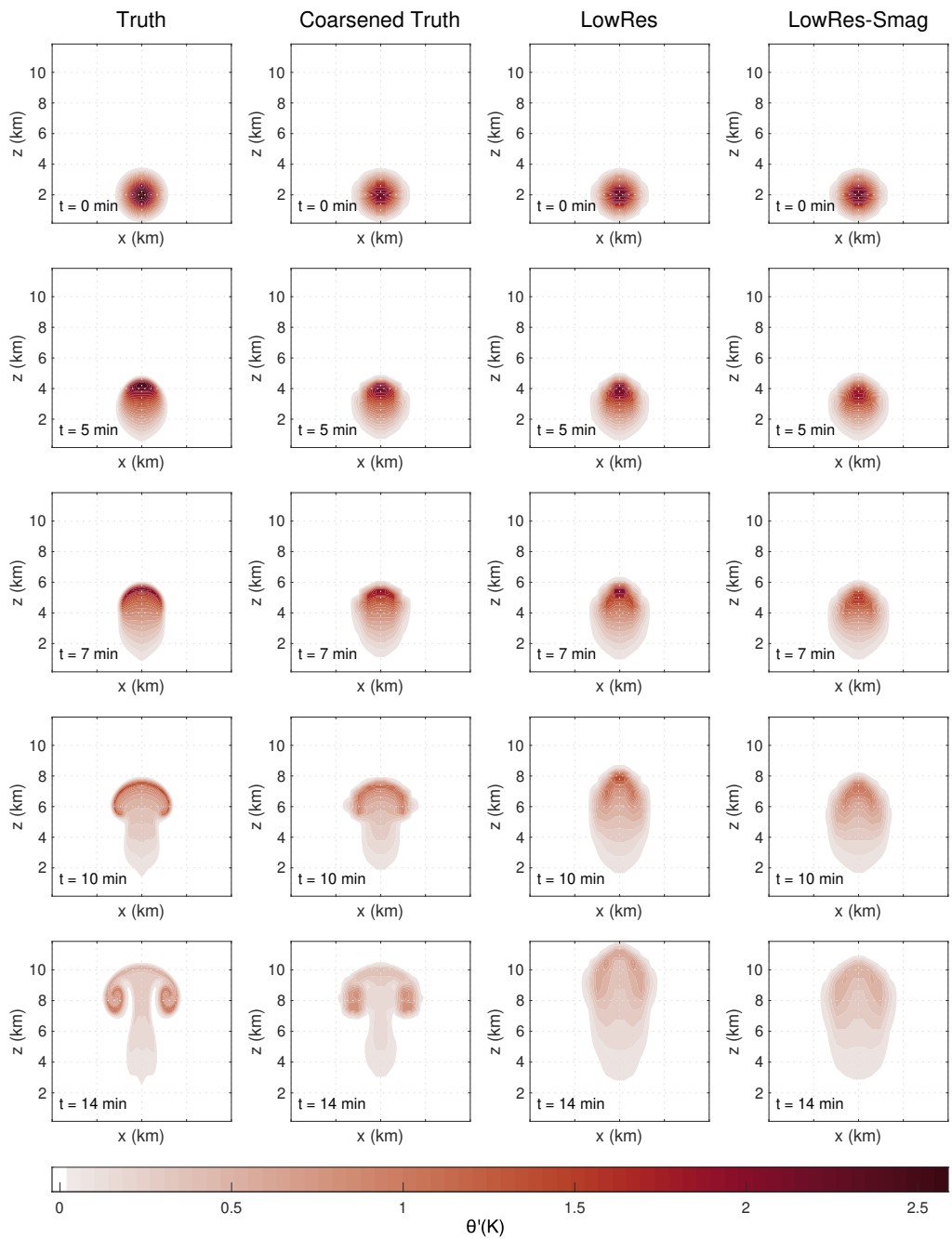

**Figure 4.** Snapshots of simulated potential temperature perturbations ($\theta'$) at t = 0, 5, 7, 10, and 14 min, with $\theta_c = 2.6K$, where the first column is the 'Truth' simulation with a high resolution of $100 \times 100 \times 100 m$, the second column is coarse-grained from the Truth simulation with a coarse resolution of $600 \times 600 \times 300 m$, the third column is the numerical simulation results with the coarse grids, and the forth column is the LowRes simulation with the Smagorinsky scheme to deal with the SGS turbulence.

This section first tests the reliability of the classic Smagorinsky scheme in the gray zone. The following testing simulations are run with LEX for comparisons: (1) a 'Truth' simulation with a high resolution of $100 \times 100 \times 100$ m as the referenced ground truth; (2) the 'Coarsened Truth', which is coarse-grained from the 'Truth' simulations, to serve as the baseline on the coarse grids; (3) a 'LowRes' simulation which is run on the coarse grids with the resolution of $600 \times 600 \times 300$ m; and (4) the 'LowRes-Smag' simulation in which the conventional Smagorinsky scheme (Smagorinsky, 1963; Shi et al., 2018) is used to solve the SGS turbulence on the coarse grids.

Figure 4 and Figure S4 clearly illustrate that the LowRes simulation tends to have a faster rising speed than the baseline simulation and it fails to resolve the correct symmetric rotor structure at the warm bubble edges due to the relatively large grid spacing. Using the Smagorinsky scheme to solve the SGS motions can slightly help correct the rising speed of the LowRes simulation. The rising speed of the thermal is lowered and is adjusted to be similar to the referenced truth state with the Smagorinsky scheme. However, the expected symmetric rotor structure cannot be simulated properly, and the intensity of the large-scale motion is also wrongly estimated, underestimation in this case. This shows that the conventional parameterization schemes still have limitations in approximating the appropriate physical dynamics and providing reliable numerical predictions in the gray zone. Therefore, improved parameterization schemes need to be developed to handle the SGS motions more precisely in such grid spacings.

## 4.2 Deep Learning-based SGS Model

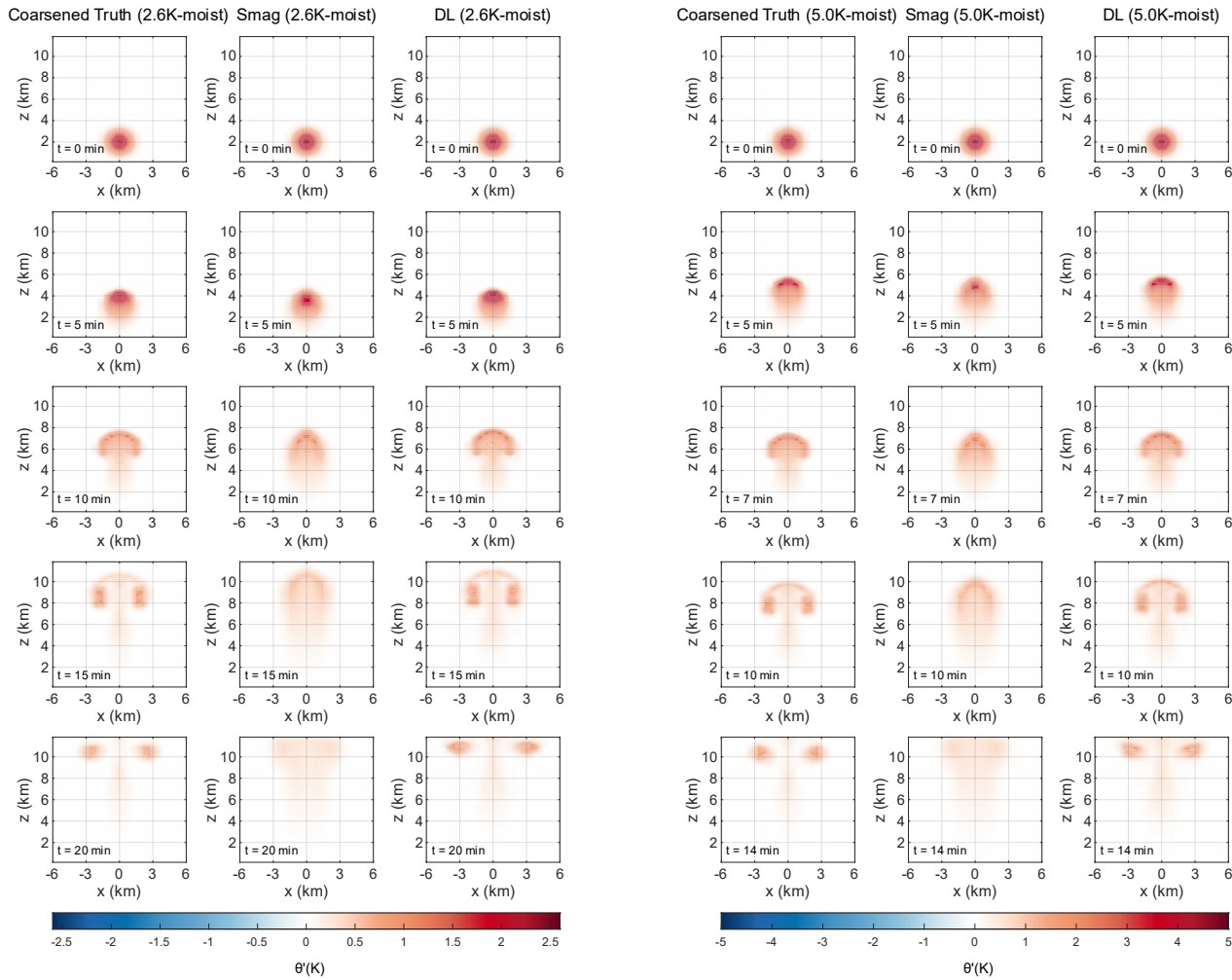

**Figure 5.** Snapshots of simulated potential temperature perturbations ($\theta'$) at t = 0, 5, 10, 15, and 20 min, with $\theta_c = 2.6K$ (the first to the third columns), and $\theta_c = 5.0K$ (the fourth to the sixth columns), where the first and fourth columns are the 'Coarsened Truth' simulations with a coarse resolution of $600 \times 600 \times 300m$, the second and fifth columns are the 'LowRes-Smag' simulations with the Smagorinsky scheme to deal with the SGS turbulence, and the third and sixth columns are 'LowRes-DL' simulations with the trained AE model to serve as the turbulence parameterization scheme.

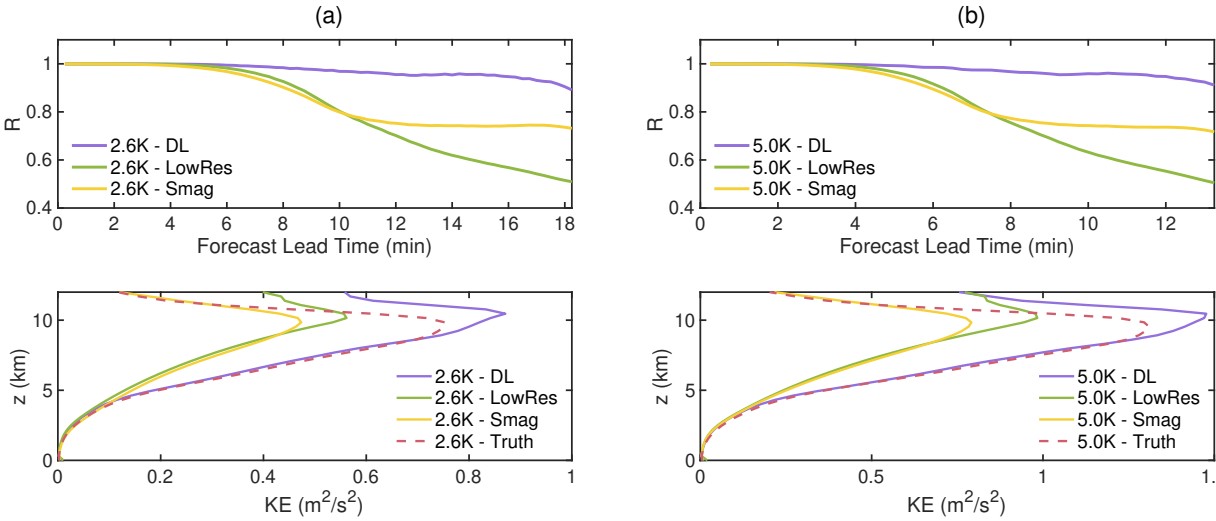

**Figure 6.** The correlation coefficient ($R$) and energy profile of the Coarsened Truth, LowRes, LowRes-Smag and LowRes-DL simulations, with $\theta_c = 2.6K$ (a), and $\theta_c = 5.0K$ (b).

The training for the DL-based SGS model is conducted with a moist warm bubble case. The online testing results are shown in Figure 5 and Figure S5. The LowRes-DL is the result of the coarse simulation with the trained AE model's correction for SGS tendencies. Compared to the conventional Smagorinsky scheme, the trained DL model is able to simulate the right rising speed, and can further develop the proper symmetric structure of the thermal, showing its superiority for turbulence predictions in the gray zone. Moreover, results are similar no matter whether the initial potential temperature is within the range of the training dataset or out of that, indicating the good generalization capability of this AE model.

The quantitative assessments of the DL model's forecast performance are also conducted with the correlation coefficient ($R$) and the kinetic energy (KE) profile, which are defined as:

$$R = \frac{\sum_i \sum_j \sum_k (X_{ijk} - \overline{X})(Y_{ijk} - \overline{Y})}{\sqrt{(\sum_i \sum_j \sum_k (X_{ijk} - \overline{X})^2)(\sum_i \sum_j \sum_k (Y_{ijk} - \overline{Y})^2)}}, \tag{27}$$

$$\text{KE} = \frac{1}{2} \langle u_i' u_i' \rangle_t, \tag{28}$$

where $X$ represents the simulated results, $Y$ represents the truth states, and the overline denotes the spatial average over all grid points for different variables. $\langle \cdot \rangle_t$ represents the time average, and $u_i' u_i'$ follows the Einstein summation convention, which equals $u'^2 + v'^2 + w'^2$.

As is seen in Figure 6, the numerical simulation with the DL-based SGS model can maintain a high level of correlation with the baseline (Coarsened Truth) during the applicable testing period. What's more, compared to the LowRes and LowRes-Smag prediction results, LowRes-DL can better forecast the small-scale turbulence motions, which is proved by the highly aligned maximum peak height of the kinetic energy and the corresponding magnitude with those of the baseline.

In this section, the experiment of the moist warm bubble case proved the capability of LEX to be used for training a DL SGS model in a physics-DL hybrid framework. The newly developed DL model can well represent SGS motions in the gray zone, offering a promising alternative to conventional parameterization schemes.

## 4.3 Comparisons and Potential Physical Insights

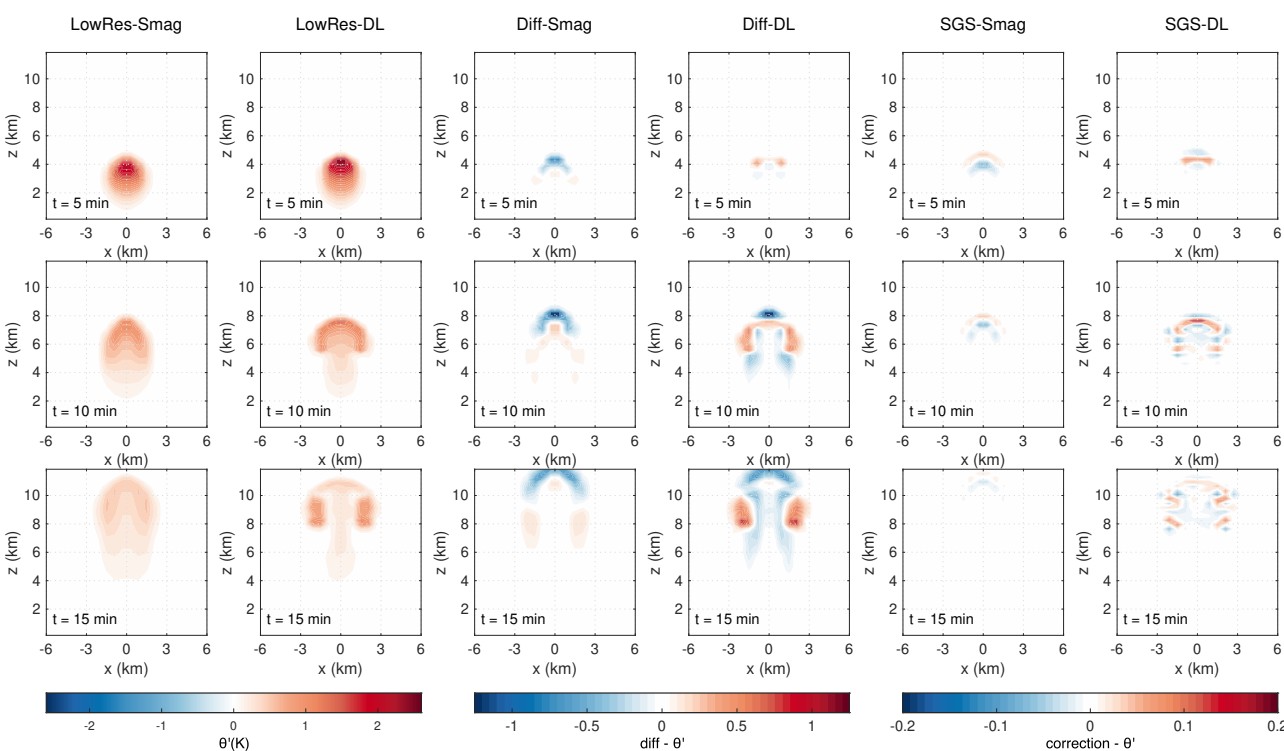

**Figure 7.** The simulated potential temperature perturbation ($\theta'$) at t = 5, 10, and 15 min, with $\theta_c = 2.6K$. The first and second columns are the forecasts of the conventional Smagorinsky and DL-based SGS model. The third and fourth columns are the differences between parameterized and non-parameterized simulation results of the conventional Smagorinsky and the DL-based SGS model. The fifth column is the SGS tendency due to the Smagorinsky model, and the sixth column is the SGS correction due to the DL model.

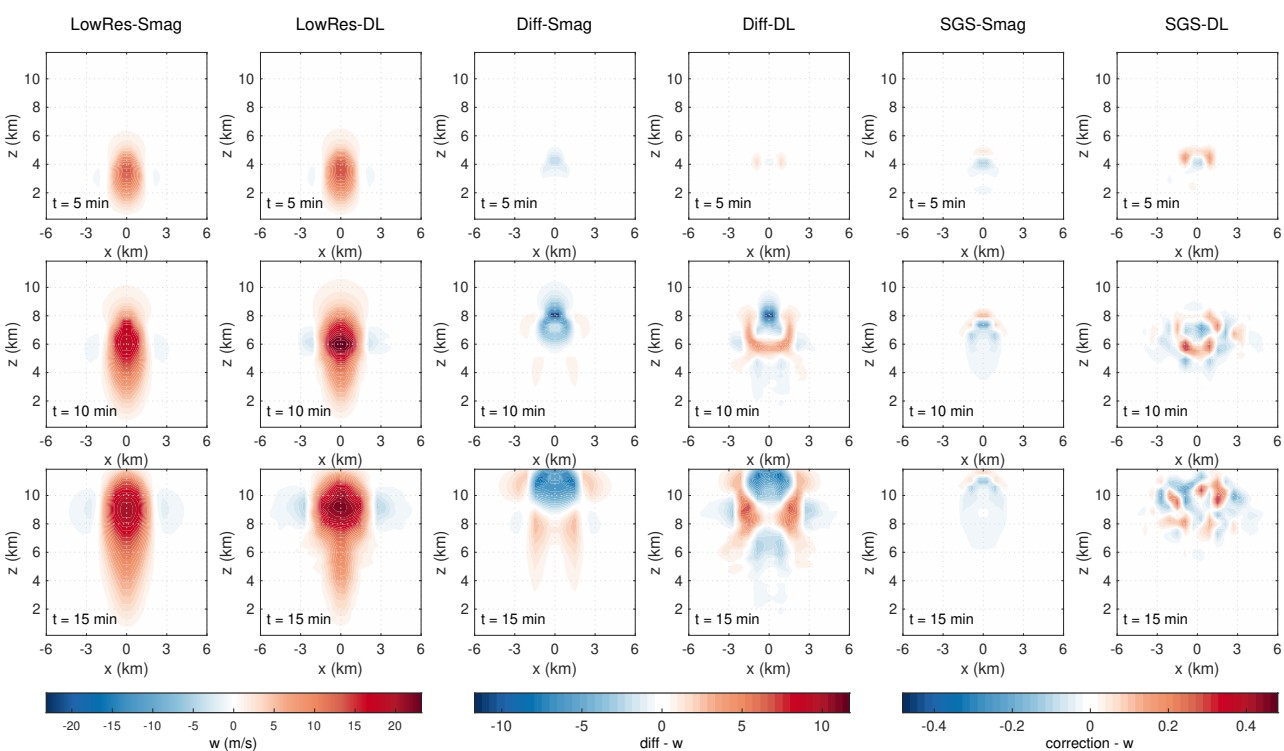

**Figure 8.** The simulated vertical velocity ($w$) at t = 5, 10, and 15 min, with $\theta_c = 2.6K$. The first and second columns are the forecasts of the conventional Smagorinsky and DL-based SGS model. The third and fourth columns are the differences between parameterized and non-parameterized simulation results of the conventional Smagorinsky and the DL-based SGS model. The fifth column is the SGS tendency due to the Smagorinsky model, and the sixth column is the SGS correction due to the DL model.

In this section, the predictions using the Smagorinsky scheme and those using the AE model against the same benchmark: the coarse-grained, high-fidelity simulation, which is the non-parameterized reference, are compared. Also the SGS tendency due to the Smagorinsky model and SGS corrections generated by the AE model are analyzed, aiming to find the potential reasons that the conventional Smagorinsky scheme fails to develop the correct rotor structure, and the difference that the hybrid model has brought. Through this way, we hope to give some physical insights from the DL-based SGS model and make some contributions to the development of the interpretable DL.

The warm bubble case is set with an initial temperature perturbation, which causes an upward buoyancy and thus gives the bubble a vertical acceleration. When the bubble rises, the cold air on each side needs to descend for compensation, which will then cause vertical velocity gradients and further form strong velocity shear layers at the bubble boundaries. In the shear layers, according to the vorticity equation, vorticity will thus be generated due to the spatial gradients of potential temperature, which is $\nabla \theta$. But when the resolution becomes coarse, small-scale processes and some key physics information, such as temperature gradients, cannot be appropriately resolved, and this causes the LowRes simulations to be unable to generate the rotor structure.

Figure 7 and Figure 8 present the forecasted potential temperature perturbation and vertical velocity from the conventional Smagorinsky scheme and the AE model, respectively. The forecast differences induced by each scheme, and the corresponding SGS tendency generated by the calssic Smagorinsky and SGS corrections generated by the AE model are also shown. Results for the additional physical quantities ($u$, $v$, and $q_v$) are provided in the supplement (Figure S6, S7, and S8).

As evidenced by Figure 7 and Figure 8, the Smagorinsky scheme and the AE model exhibit obviously different impacts on the development of the warm bubble at the very beginning. Compared with the coarse-grained high fidelity simulations (the third and fourth column in Figure 7 and Figure 8), the Smagorinsky mainly imposes a cooling effect on the warm bubble, and weakens its upward motion. But the AE model sustains warming and the upward motion in regions that are to further develop the rotor structure. This significant difference is key to the later development of the warm bubble.

A comparative analysis of the SGS tendency due to the Smagorinsky model and the corrections due to the AE model provides further explanations (see the fifth and sixth columns in Figure 7). As the conventional Smagorinsky is a diffusion model, it naturally diffuses warm temperature anomaly to surrounding regions. Accordingly, it produces a warming tendency near the top of the rising thermal, a cooling tendency below, but almost no extra effect at the rising centre, which leads to the dissipation of the thermal's original energy without any replenishment. However, the trained AE model can accurately produce a warming correction at the thermal centre, and thus help maintain the buoyancy force that drives the bubble's sustaining development.

The above findings can explain Figure 4, where the classic Smagorinsky helps correct the warm bubble's rising speed compared to the LowRes results, as the Smagorinsky scheme greatly lowers down the temperature at the top with its energy diffused at the thermal centre. Furthermore, they align with Figure 6, where the Smagorinsky forecast has the same energy and vertical velocity peak with the hybrid model, but it presents smaller values, even smaller energy than the LowRes simulation.

The classic Smagorinsky tends to produce overly diffusive corrections, which limits it to resolve fine-scale structures and maintain the necessary energy for the warm bubble to develop the rotor structure. The corrections generated by the AE model are much more detailed and accurate. As is illustrated in Figure 7, SGS corrections of the AE model always help maintain the strength of the potential temperature at the critical part of the warm bubble in a very fine way, such as the rising centre

**Table 1.** Computational performance comparison of CM1 and LEX. The CM1 was run using 64 cores of an AMD Ryzen Threadripper 3990X. The LEX was run on NVIDIA RTX A6000 GPU. No SGS models were used in CM1 or LEX simulations.

| Model | Resolution (m) | Time Step (s) | Hardware (CPU/GPU) | Integration Time (min) | IO/Setup and Compilation Time (s) | Execution Time (s) |
|---|---|---|---|---|---|---|
| *CM1* | 100×100×100 | 2 | CPU (64 cores) | 20 | 40.00 | 789.00 |
| *LEX* | 100×100×100 | 2 | GPU | 20 | 171.97 | 548.01 |
| | 100×100×100 | 12 | GPU | 20 | 171.97 | 84.75 |

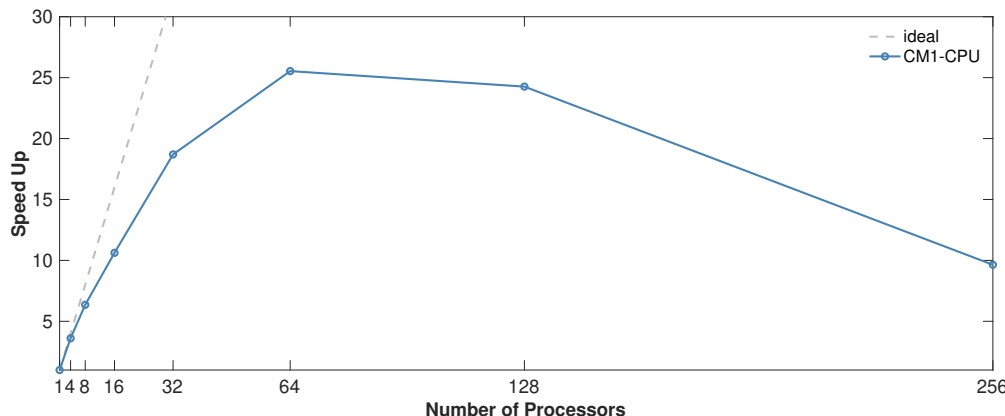

**Figure 9.** Strong scaling performance of CM1 on the AMD Ryzen Threadripper 3990X.

at the key beginning, and the rotors on the sides after they have been maturely developed. This makes the hybrid model keep the energy for rising and developing the rotors. These detailed structures are probably essential to enable the model to model the small-scale physics information which is unresolvable by the coarse grid. Similarly, Figure 8 shows that the AE model's corrections exhibit detailed structures and help keep the upward motion.

## 5 Computing Time Comparisons

### 5.1 LEX Compared to CM1

The computational costs are compared in this section. As mentioned in Section 1, LEX has better numerical stability and is expected to show faster computing speed with JAX acceleration techniques. Using the conventional CM1 model as the benchmark model, Table 1 shows that employing the same time step of two seconds to run a 20-minute simulation, the total computing time for CM1 is 789 s using 64 cores, while the LEX run takes 548 s on one GPU. Furthermore, at the resolution of $100 \times 100 \times 100$ m, the longest time step for CM1 to maintain numerical stability is two seconds, but for LEX, it can be

**Table 2.** Computational speed comparison of DL-based SGS model and conventional Smagorinsky Scheme, with the resolution being $600 \times 600 \times 300\ m$, and the 15-second time step for a 20-minute simulation test for each.

| Model | Hardware | Precision | Parameterization Scheme | IO/Setup and Compilation Time(s) | Execution Time (s) | Model Inference Time (s) |
|---|---|---|---|---|---|---|
| *LEX* | GPU | fp32 | N/A | $\sim 36$ | 0.89 | N/A |
| *LEX+Smag* | CPU | fp32 | Smagorinsky | $\sim 33$ | 43.28 | N/A |
| *LEX+Smag* | GPU | fp32 | Smagorinsky | $\sim 33$ | 1.91 | 1.02 |
| *LEX+DL* | GPU | fp32 | DL | $\sim 65$ | 1.48 | 0.59 |
| *LEX+DL* | GPU | fp64 | DL | $\sim 65$ | 6.18 | N/A |

up to twelve seconds, thanks to its acoustic-wave-filtering equations and the strong stability integration scheme SSPRK3. As a result, LEX's running time can be further reduced by a factor of 1/6. Meanwhile, according to the strong scaling test shown in Figure 9, the speed-up factor for the 20-minute simulation of CM1 reaches the maximum with 64 processors. That means in this 20-minute simulation for the warm bubble case, compared to the optimal speed-up performance of CM1 with 64 CPU cores, LEX on a single GPU is around nine times faster.

Because the 20-minute simulation is a relatively short integration period, leading to the LEX setup and just-in-time compilation time accounting for a significant fraction of the total running time. However, if we run the LEX for a substantially longer time, the compilation and setup time probably can be ignored. This demonstrates the great application potential of LEX to run for long simulations.

The effectiveness of GPU acceleration is also shown in Table 2. Calculating with the same resolution and a 15-second time step for a 20-minute integration time, LEX with the Smagorinsky scheme runs around 21 times faster on the GPU than on the CPU, excluding the just-in-time compilation time.

## 5.2 DL-based SGS Model Compared to Conventional Smagorinsky Scheme

LEX can be trained with a DL-based SGS model and succeed in numerical predictions in the gray zone, but whether such physics-DL hybrid models can be applied in real weather forecasts also relies on their computational costs. The parameterizations for SGS processes are only one part of the entire numerical weather predictions, thus, they are expected to run at a fast speed. Since the DL model is trained with the double-precision float64, its computing time is first evaluated with the same precision to run the hybrid model. Table 2 shows that when running with float64, the LEX-DL model with a 15-second time step takes around three times of the computing time of the LEX-Smag model using float32 with a same time step after compilation, and meanwhile its compilation time is two times slower, which is not satisfying performance. One reason for this is that float64 needs more computational resources than float32, and the other is the hardware limitation that further increases the computational costs, as float64 convolutions are not supported by XLA on the NVIDIA RTX A6000 GPU now, which is used in the model evaluations for this paper.

However, though the double precision is necessary for the training of the LEX-DL model, a single-precision of float32 is found to be applicable for the evaluations, as the model parameters have already been sufficiently trained and the DL model will not cause any tiny noise towards the stable thermal structure. Thus, the computing efficiency of the DL-based SGS model is further enhanced. As shown in Table 2, using the same time step of 15 seconds, the LEX-DL model with a single precision can achieve $76\%$ computing time reduction than that with the double precision, which only needs $1.48s$ to complete the integration task after the compilation.

We also test the Smagorinsky model and the AE model on CPUs and it turns out that the Smagorinsky model will be around 30% faster. But if we conduct the test on GPUs, it is found that though the compilation time is two times slower, the fastest speed the hybrid model can achieve using GPUs now after compilation is comparable to that of the LEX-Smag model with the single precision, which means the DL model can enable a lower computational expense for prolonged forecasts.

## 6 Conclusions

As the model resolutions are entering the kilometer-scale range, parameterizations for SGS motions in the gray zone remain key obstacles in today's numerical weather forecasts, because turbulence and convection can only be partially resolved and conventional parameterization schemes are no longer applicable in the gray zone. LES models are always valuable and important tools for studying small-scale turbulence motions in the field of atmospheric science. They are used to compare the different SGS parameterization schemes and help develop improved SGS models for different flows (Remmler and Hickel, 2013; Khani and Waite, 2015). However, LES that is available for large domains is still lacking to date. In this background, the new LES model written with JAX, LEX, is developed in this paper. By validating its simulation results with those of the traditional CM1 model using different initial conditions for a simple thermal case, LEX is proven to be a reliable and robust LES model.

Moreover, LEX can be applied for simulations on large domains with its fast computation speed. With GPU acceleration, the acceleration tools from JAX, and good numerical stability that allows larger time steps, running LEX on one GPU is as fast as running CM1 on 600 CPU cores. One disadvantage of LEX is that the just-in-time compilation takes much time. Therefore, LEX is better used for long-period simulations. As the integration time increases, the advantage of LEX's fast computational speed will become increasingly apparent, compared to the other traditional LES models.

The newly developed LEX code is also auto-differentiable. To report its differentiability, based on LEX, a DL-based SGS model is further trained for SGS parameterization in the gray zone for the thermal bubble case. A simple AE model is applied to produce correction terms for the prognostic variables. The coupled online training of the physics-DL hybrid model integrates the dynamics in the loops every epoch. The trained model exhibits excellent capability to correct the dynamical core integration and simulate the symmetric rotor structure of the rising thermal in the gray zone. The traditional Smagorinsky scheme is also tested and exhibits poorer performance, with the thermal perturbation wrongly estimated and failing to produce the rotor structure.

The DL model is not only more reliable in representing the SGS turbulence in the gray zone, but its inference time can also be comparable to that of the conventional parameterization scheme. The preliminary results show that although the training process requires double precision which will lead to great computational costs, the trained model is able to be run with single precision, enabling even faster computing speed than the classic Smagorinsky after compilation. However, the hybrid model needs longer compilation time, and makes its total computing time twice that of the classic Smagorinsky for the 20-minute simulation in this case. These indicate that hybrid models are promising tools to be applied for SGS representations for real atmospheric forecasts, especially for prolonged simulations which can significantly reduce the proportion of compilation time.

LEX v1.6.0 is the initial model that has completed the initial accuracy tests and been validated for the hybrid model training. However, now it is still an idealized model which does not contain the microphysics and radiation scheme. This LEX version has already included a surface flux scheme following Neale et al. (2010), though it was not tested in this study. The implementation of P3 microphysics (Milbrandt et al., 2021) is ongoing and will be tested in cloudy boundary layer cases. We will also add the Rapid Radiative Transfer Model (RRTMG) (Iacono et al., 2008) to LEX in the future.

Moreover, LEX is designed with inherent support for parallelism thanks to its implementation using the JAX framework, which provides automatic parallelization capabilities with the Single-Program Multi-Data (SPMD) codes (Bradbury et al., 2018), as well as mpi4jax (Häfner et al., 2021). However, LEX v1.6.0 applies the BiCGSTAB algorithm as the pressure solver, which introduces per-iteration global synchronization points that are conflict with the subdomain-level synchronization needed for ghost-point exchanges. Considering the problem scale and hardware setup, the SPMD parallelism across the spatial domain is considered not plausible now, and mpi4jax is the appropriate tool for domain decomposition and parallel computing. Now the parallelism of LEX v1.6.0 can only be performed at the batch level. Related codes are provided on GitHub.

Overall, LEX v1.6.0 can now be utilized with accuracy and fast-computing speed. It is auto-differentiable so that corresponding DL-based SGS models can be trained to provide high-fidelity parameterizations for SGS motions in the gray zone. The development of LEX is expected to help deepen knowledge of the small-scale turbulence processes and enable the future development of more reliable parameterization schemes in the gray zone.

*Code availability.* The current version of LEX is publicly available on Github at https://github.com/MetLab-HKUST/LEX under the MIT license. LEX codes, and scripts for producing figures are archived on Zenodo under https://doi.org/10.5281/zenodo.15486687 (Zhu et al., 2025a). Related data used in this study can be accessed from Zenodo under https://doi.org/10.5281/zenodo.15730773 (Zhu et al., 2025b). CM1 version cm1r21.1 is used in this study, which can be accessed from https://github.com/NCAR/CM1.

*Author contributions.* SXM proposed this study and developed LEX. ZXY developed the dataset, conducted the preliminary training and evaluations. ZXY, QYQ, and SXM contributed to the paper writing together.

*Competing interests.* The authors have no competing interests to declare.

*Acknowledgements.* The work is substantially supported by a grant from the Research Grants Council (RGC) of the Hong Kong Special Administrative Region, China (Project Reference Number: AoE/P-601/23-N). Additionally, XZ and XS also received support from the RGC Grant HKUST-16301322 and the Center for Ocean Research in Hong Kong and Macau (CORE), a joint research center between the Laoshan
Laboratory and the Hong Kong University of Science and Technology (HKUST). YQ acknowledges funding from NSF through the Learning the Earth with Artificial intelligence and Physics (LEAP) Science and Technology Center (STC) (Award #2019625).

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
