# Peer review of "LEX v1.6.0: A New Large-Eddy Simulation Model in JAX with GPU Acceleration and Automatic Differentiation"

_EGUsphere, 2025_

## Referee Comment (RC2)

**Comments on: *LEX v1.4: A New Large-Eddy Simulation Model in JAX with GPU Acceleration and Automatic Differentiation**

egusphere-2025-2568

**General Comments**

The present work proposes a deep learning (DL)-based subgrid-scale (SGS) model for large-eddy simulations (LES) of gray zone turbulence in atmospheric flows. The authors implement a JAX-based code framework called LEX which allows end-to-end training of DL-based explicit correction terms to the grid-filtered governing equations. The DL SGS model is trained on coarse-grained data for a rising thermal. A posteriori tests show that the DL SGS model offers improved approximation quality compared to the classical Smagorinsky model.

The overall approach of the paper follows the current trend of hybrid machine learning (ML) and computational fluid dynamics (CFD). The technical novelty of the proposed method is therefore limited. However, applying hybrid ML-CFD to gray zone LES modeling is novel, and the presented results of applying DL-based SGS models to such flows are encouraging. In addition, the LEX framework appears to be a good starting point for future developments of DL-based surrogates for numerical weather prediction.

While the presentation of the manuscript is clear, the paper lacks thorough quantitative analysis and scientific precision in parts. Some claims and statements of the authors are too vague, too imprecise or lack sufficient foundation.

Therefore, the paper can only be reconsidered for publication after a major revision, in which the below listed comments are convincingly addressed.

**Specific Comments**

**Major comments**

1. There are statements throughout the manuscript which are scientifically imprecise or lack supporting evidence. For example:

   - lines 5 & 6" "Thus, developing SGS turbulence models for the gray zone requires new LES models, which ... enable new approaches to develop SGS models". I do not understand what the authors mean by this? To my understanding, an LES model already contains a SGS model. Therefore, how can a new LES model enable the development of an SGS model?

   - Line 8: "The new LES model is capable of adequate parallelism ...". How is this claim supported? To my understanding, the LEX model is only run on a single GPU, and parallel simulations are not discussed at all.

   - 21 & 22: "The capability of LESs to simulate small-scale turbulence motion...". This sentence feels scientifically imprecise. LES is supposed to resolve large-scale motions while only modeling the effect of small-scale turbulence on aforementioned large eddies.

- Line 40: "GPU codes are known to run much faster than conventional Fortran or C codes on CPUs". In my opinion, this statement is too generic and scientifically imprecise. While GPUs leverage massive thread-level parallelism, achieving actual code speed up is highly dependent on specific applications.

- Lines 68 & 69: "Existing studies have also shown that JAX-GPU codes enable ... less computational costs when the problem sizes become quite large". What do the authors mean by this? Surely, the computational cost can not decrease with increasing problem size? What does "quite large" mean?

- Line 298: "... float64 convolutions are not supported by XLA now,...". In my opinion, this is not true. float64 support is backened (i.e. hardware) specific. The NVIDIA A6000 GPU does in fact not natively support float64. However, NVIDIA A100 or H100 GPUs provide float64 support on the hardware side. Please correct this statement.

2. What is the motivation to choose the conventional Smagorinsky model as a baseline for comparison? It is well known that the dynamic Smagorinsky model outperforms the classical model in many scenarios. This would be a much stronger baseline for benchmarking the DL-based SGS model.

3. I have the following comments and questions regarding model training:

- What is the time step size of the coarse-grained simulation? The time step size of the high-resolution simulation is 5s. While the spatial coarse-graining factors are explicitly mentioned, the authors do not mention whether temporal coarse-graining is also applied.

- What is the rationale for choosing a 6x CG in the horizontal direction and a 3x CG in vertical direction?

- Please specify the loss functional explicitly. Specifically, the mean-squared error of which quantities is used?

- Please provide more information regarding hyperparameters of the model training. How many optimization steps are used during training? What is the final training loss level? What is the stopping criterion? What is the learning rate? Is there a learning rate scheduler?

- The authors mention, that the training for the dry case can "achieve asymptotic convergence" while it "shows oscillatory convergence behavior" for the moist case. Please add loss plots for both scenarios to the manuscript (e.g., to the appendix).

- Are the authors using custom implementations to propagate the AD gradients through the BiCGSTAB solve?

4. I have the following comments and questions regarding the chosen parameterization and DL model:

- The standard WENO3- and WENO5-JS schemes are known to be overly dissipative. What is the motivation to choose this parameterization?

- The DL-SGS model output is applied to $\theta, u, v, w$ and $q_v$. Does the same hold true for the Smagorinsky model? Is the mixing ratio of water vapor $q_v$ a transported quantity or is it post-processed?

5. Validation of the LEX solver with CM1 results are purely qualitative. Please add quantitative comparisons if possible. The authors mention that "results of the LEX are identical with those of CM1". This is an overstatement in my opinion, as Fig. 2 shows visible discrepancies between the two simulations. For example, the lower parts of the thermal are clearly different at later times and the structure of the rotors show differences. The authors should tone down this claim or provide quantitative evidence for it.

6. I would encourage the authors to verify the implementation of the AD gradients with finite-difference analogs. A simple test case can be chosen in which only a single parameter of the DL-based model is optimized with AD and with FD. The error between the two should converge as the step size in the FD approximation approaches zero.

7. The authors mention the trained DL model can "develop the proper symmetric structure of the thermal". This statement is not true. Figs. 4 and 6 clearly show that the DL model breaks symmetry. The authors themselves acknowledge this fact later in Section 4.2.2.

8. It is mentioned that the mixing ratio of water vapor has to be clipped after application of the DL model (Section 2.2.2). I am interested how often this occurs for the trained model over the course of a simulation.

9. I agree with the authors that the DL-based SGS model outperforms the conventional Smagorinsky model for the thermal test case. To my understanding, the DL model is applied after a full integration step while the Smagorinsky model is applied per stage (i.e., thrice per integration step). Can the authors elaborate on this? It would be very interesting to visualize the output of the DL model to try to understand its improved SGS modeling capabilities. Have the authors done such analyses? Is the model output interpretable? What conclusions can be drawn from it?

10. I have the following comments and questions regarding the computing time comparison:

    - The performance comparison in Section 5 is somewhat misleading. The authors claim that they achieve a 92:1 speed up when comparing the LEX code run on an A6000 GPU with the CM1 code run on a single CPU core. I think the authors are aware that such a comparison is not meaningful at all. Can the authors comment on this?
    - In Section 5.2, the wall-clock time of the DL-based SGS model is compared with the Smagorinsky model. Given the short simulation time, the wall-clock time measurements are strongly influenced by the duration of the just-in-time compilation. I would encourage the authors to simply evaluate the Smagorinsky model and the DL-based SGS model on their own to provide more meaningful WCT measurments or to exclude the duration of the jit-compilation from the performance measurements.

11. The authors should consider citing JAX-Fluids [1, 2] and [3]. JAX-Fluids is a JAX-based fully-differentiable CFD solver for compressible single- and two-phase flows, which is closely connected with the present research. Specifically, JAX-Fluids implements functionality for LES and has been used for end-to-end training of implicit LES models [3].

**Minor comments**

1. What is the reason for v1.4 in the title of the manuscript? Maybe I have missed it, but it is not mentioned in the remainder of the paper. Is the present work building upon a previous release of the LEX solver?

2. In section 2.1.1, some variables are not defined, including $\epsilon, c_p, c_v, w, g, p_s, R$. While I assume that many of these quantities are well known (presumably, $c_p$ is the heat capacity at constant pressure), it would improve clarity to specify their definition once.

3. Please define the correlation coefficient $R$ and the kinetic energy $KE$ in Section 4.2.

**Technical Corrections**

1. Please proofread and type-check the manuscript carefully. A couple of typos:

    (a) *In the gray zone, turbulence and convection ...* in line 30.

(b) *the acoustic-wave-filtered equations ... are adopted* in line 84.

(c) *for validation simulations.* in lines 178 & 179.

(d) I think the abbreviation *LESs* is not commonly used.

**References**

[1] D. A. Bezgin, A. B. Buhendwa, N. A. Adams, Jax-fluids: A fully-differentiable high-order computational fluid dynamics solver for compressible two-phase flows, Computer Physics Communications 282 (2023) 108527. `doi:10.1016/j.cpc.2022.108527`.
URL `https://linkinghub.elsevier.com/retrieve/pii/S0010465522002466`

[2] D. A. Bezgin, A. B. Buhendwa, N. A. Adams, Jax-fluids 2.0: Towards hpc for differentiable cfd of compressible two-phase flows, Computer Physics Communications 308 (2025) 109433. `doi:10.1016/j.cpc.2024.109433`.
URL `https://linkinghub.elsevier.com/retrieve/pii/S0010465524003564`

[3] D. A. Bezgin, A. B. Buhendwa, S. J. Schmidt, N. A. Adams, Ml-iles: End-to-end optimization of data-driven high-order godunov-type finite-volume schemes for compressible homogeneous isotropic turbulence, Journal of Computational Physics 522 (2 2025). `doi:10.1016/j.jcp.2024.113560`.

---

## Author Comment (AC2)

**The Response to Reviews of**

**LEX v1.4: A New Large-Eddy Simulation Model in JAX with GPU Acceleration and Automatic Differentiation**

Xingyu Zhu, Yongquan Qu, Xiaoming Shi, Geoscientific Model Development (GMD)

Original submission reference #: EGUSPHERE-2025-2568

**Dear Editor and Reviewers:**

This is a response letter and an updated version of our manuscript to *GMD*. The original version was submitted in July, 2025 and used a CNN for the hybrid approach training. In the submitted manuscript, we pointed out some existing problems to be solved, and the reviewers also showed much concern about them. During the revision process, we made a major breakthrough, solved most of the obstacles, and successfully improved our results by training with a new autoencoder (AE) model, and a newly designed loss function. Thus, this revised manuscript has been almostly substantially rewritten to present our up-to-date research progress.

An AE model has replaced the initial CNN model to be coupled with the LEX model for training. The AE model has less model parameters, enabling faster computing speed and more look-ahead steps during training than the initial CNN. By using the newly designed loss function which contains an L2 loss, a Laplacian loss, and a term to penalize the negative values of water vapor produced by the DL model, the trained model shows more steady training process without any manual intervention, like clipping the negative values of watervapor, and better application results. Now the forecast results of the hybrid model are strictly symmetric without any background noise caused by the DL model for the moist case, and the time step is further extended to 15 seconds. Moreover, single precision can be applied to the entire hybrid model for validations now. Thus, the related part of the dry case has been deleted. Data and analyses regarding the computational cost comparisons have also been re-tested and rewritten.

Thanks to the reviewers, the SGS corrections have also been plotted out and analyzed, bringing some physical insights to this manuscript. We have also revised other parts of the manuscript in response to the reviewers' comments, which has significantly improved its overall quality. Our detailed responses to the comments are provided in the following pages.

We sincerely thank the editor and reviewers for their thoughtful feedback and for considering our revised manuscript.

Sincerely,

Xingyu Zhu, Yongquan Qu, and Xiaoming Shi

Division of Environment and Sustainability
The Hong Kong University of Science and Technology

**1 Response to Reviewer 1**

**Comment 1.1**

Line 57: There is a listing of the advantages of the hybrid machine learning approach to atmospheric sciences, but weather forecast accuracy is in my opinion not one of them. Purely data-driven DL models, such as Google's graphcast and ECMWF's AIFS nowadays exceed forecast accuracy of both physical and hybrid-ML models. So I would recommend to remove the 'accurate weather predictions' from this line.

**Response:**

Thanks a lot for this suggestion. We realize this description is not very rigorous and have removed it.

Manuscript text (Lines 56-58):

" By integrating the physics-based core directly into the training loop, these hybrid approaches tend to yield more reliable and interpretable weather and climate predictions than purely datadriven DL models."

**Comment 1.2**

In line 195 the authors explain that the accumulated MSE forms the basis of the loss calculation for training the SGS. Are the authors concerned with a 'smearing' effect of the SGS tendency, could this be observed?

**Response:**

Thank you for this insightful question. To deal with the 'smearing effect', we have now updated our loss functions used for the training of the hybrid model. The updated loss functions mainly contain three terms: the L2 loss, the term used to penalize the negative values of water vapor produced by the DL model, and a Laplacian loss. The Laplacian loss, which is written as:  $\mathcal{L}_{Lap} = \|\nabla^2 M^k x_{t_0} - \nabla^2 x_{t_k}\|_2^2, \text{ helps prevent over smoothing and enhance details with the second-order differential operator. With these loss terms applied, now the trained model can give quite good performance with the predicted rotor structure being strictly symmetric (see Figure 5 and Figure S5). The updated loss function, related explanations, and figures are shown below.$

**Reference:**

Li, S., Xu, X., Nie, L., and Chua, T.-S.: Laplacian-steered neural style transfer, in: Proceedings of the 25th ACM international conference on Multimedia, p. 1716–1724, https://doi.org/10.1145/3123266.3123425, 2017.

Didwania, K., Gakhar, I., Arya, P., and Labroo, S.: LapLoss: laplacian pyramid-based multiscale loss for image translation, https://doi.org/2503.05974, 2025.

Figure 5: Snapshots of simulated potential temperature perturbations ( $\theta'$ ) at t = 0, 5, 10, 15, and 20 min, with  $\theta_c = 2.6K$  (the first to the third columns), and  $\theta_c = 5.0K$  (the fourth to the sixth columns), where the first and fourth columns are the 'Coarsened Truth' simulations with a coarse resolution of  $600 \times 600 \times 300m$ , the second and fifth columns are the 'LowRes-Smag' simulations with the Smagorinsky scheme to deal with the SGS turbulence, and the third and sixth columns are 'LowRes-DL' simulations with the trained AE model to serve as the turbulence parameterization scheme.

Figure S 5: Snapshots of simulated water vapor mixing ratio  $(q_v)$  at t = 0, 5, 10, 15, and 20 min for moist cases, with  $\theta_c = 2.6K$  (the first to the third columns), and  $\theta_c = 5.0K$  (the fourth to the sixth columns), where the first and fourth columns are the 'Coarsened Truth' simulations with a coarse resolution of  $600 \times 600 \times 300m$ , the second and fifth columns are the 'LowRes-Smag' simulations with the Smagorinsky scheme to deal with the SGS turbulence, and the third and sixth columns are 'LowRes-DL' simulations with the trained AE model to serve as the turbulence parameterization scheme.

**Manuscript text (Lines 205-238):**

"At each step, the  $L_2$  loss, Laplacian loss, as well as an extra loss term which is used to penalized unreasonable model outputs are employed and accumulated to be the total loss of the current training step, with which we use the Adam (Kingma and Ba, 2017) optimizer to adjust the DL model parameters. In this study N=12. To mitigate the influence of the potential rounding errors, double-precision (float64) is employed throughout the training process.

The  $L_2$  loss is written as:

$$\mathcal{L}_{l_2} = \frac{1}{2} \| \boldsymbol{M}^k \boldsymbol{x}_{t_0} - \boldsymbol{x}_{t_k} \|_2^2,$$

where x is the tensor of the general physical state consisting of density potential temperature perturbation ( $\theta'$ ), mixing ratio of water vapor perturbation (qv'), horizontal and vertical velocity (u, v, w). M represents the LEX model (dynamical core and the SGS model, if any),  $x_{t_0}$  is the initial state of x, and  $x_{t_k}$  is the truth state of x at the  $k^{th}$  look-ahead step during training.

When training, it is found that the DL model can not distinguish the physical meaning of the input variables and will generate unreasonable outputs in the very beginning, such as negative values for water vapor, and meaningless background noise, which are against the laws of physics and will strongly impact the integrated results of the numerical simulation in the next time step during the training loop.

Thus, firstly, to penalize such unreasonalbe negative values that may be caused by the DL model, an extra loss term is added, which is:

$$\mathcal{L}_{penal-neg} = \begin{cases} -q_v, & \text{if } q_v

Figure 9: Strong scaling performance of CM1 on the AMD Ryzen Threadripper 3990X.

**Table 2.** Computational speed comparison of DL-based SGS model and conventional Smagorinsky Scheme, with the resolution being  $600 \times 600 \times 300 \ m$ , and the 15-second time step for a 20-minute simulation test for each.

| Model         | Hardware | Parameterization | IO/Setup | Compilation | Execution | Model Inference |
|---------------|----------|------------------|----------|-------------|-----------|-----------------|
|               |          | Scheme           | (s)      | (s)         | Time (s)  | Time (s)        |
| LEX           | GPU      | N/A              | ~5       | ~31         | 0.89      | N/A             |
| LEX           | CPU      | Smagorinsky      | ~5       | ~28         | 43.28     | N/A             |
| LEX           | GPU      | Smagorinsky      | ~5       | $\sim$ 28   | 1.91      | 1.02            |
| LEX+DL (fp32) | GPU      | DL               | $\sim 5$ | $\sim 60$   | 1.48      | 0.59            |
| LEX+DL (fp64) | GPU      | DL               | $\sim 5$ | $\sim 60$   | 6.18      | N/A             |

"The computational costs are compared in this section. As mentioned in Section 1, LEX has better numerical stability and is expected to show faster computing speed with JAX acceleration techniques. Using the conventional CM1 model as the benchmark model, Table 1 shows that employing the same time step of two seconds to run a 20-minute simulation, the total computing time for LEX is 789 s using 64 cores, while the LEX run takes 548 s on one GPU. Furthermore, at the resolution of  $100 \times 100 \times 100$  m, the longest time step for CM1 to maintain numerical stability is two seconds, but for LEX, it can be up to twelve seconds, thanks to its acoustic-wave-filtering equations and the strong stability integration scheme SSPRK3. As a result, LEX's running time can be further reduced by a factor of 1/6. Meanwhile, according to the strong scaling test shown in Figure 9, the speed-up factor for the 20-minute simulation of CM1 reaches the maximum with 64 processors. That means in this 20-miniute simulation for the warm bubble case, compared to the optimal speed-up performance of CM1 with 64 CPU cores, LEX on a single GPU is around nine times faster.

Because the 20-minute simulation is a relatively short integration period, leading to the LEX setup and just-in-time compilation time accounting for a significant fraction of the total running time. However, if we run the LEX for a substantially longer time, the compilation and setup time probably can be ignored. This demonstrate the great application potential of LEX to run for long simulations.

The effectiveness of GPU acceleration is also shown in Table 2. Calculating with the same resolution and a 15-second time step for a 20-minute integration time, LEX with the Smagorinsky scheme runs around 21 times faster on the GPU than on the CPU, excluding the just-in-time compilation time."

Reference: I believe a reference to "JAX-Fluids: A fully-differentiable high-order computational fluid dynamics solver for compressible two-phase flows" (https://doi.org/10.1016/j.cpc.2022.108527) should be added as it closely aligns with this work.

**Response:**

Thanks a lot, the references have been added.

Manuscript text (Lines 68-70):

"Existing work includes JAX-Fluids, a Python-based end-to-end differentiable CFD framework which is designed with JAX for compressible single and two-phase flows (Bezgin et al., 2023, 2025a), and enables end-to-end training of DL-based implicit LES models (Bezgin et al., 2025b)."

**Comment 1.7**

I believe the manuscript lacks a proper explanation of the applied boundary conditions. I suspect you are using periodic lateral boundaries and a sponge layer at the top? Please elaborate in the theoretical section.

**Response:**

It's the periodic lateral boundaries and a free-slip layer at the top that are used in this paper. Thanks for this reminder and we have declared the applied boundary conditions to the section 2.1.3 from Line 142 to Line 143.

Manuscript text (Lines 142–143):

"... Periodic boundary conditions are applied to the four sides and rigid, free-slip wall boundary conditions are specified at the top and bottom of the domain. ..."

**Comment 1.8**

I would love to see a short example of a more realistic emergent cumulus case, and especially whether the trained SGS parameterization from the warm bubble can be transferred to a cloudy atmosphere.section.

**Response:**

Thanks a lot for your interest in our work. We'd love to provide a related case for you in our upcoming work for the next version of LEX, and the implementation and testing of a cloud microphysics scheme in LEX is ongoing. However, as for this paper, it is LEX v1.6.0, which does not contain the microphysics part.

The manuscript lacks an outlook with respect to missing components in LAX: microphysics, radiation, surface scheme. It should be mentioned in the article what the status of these elements are and how this limits the applicability of LAX.

**Response:**

Thanks for this reminder. We have now added the model outlook part to the revised manuscript in the conclusion, from Line 405 to Line 409.

Manuscript text (Lines 406–410):

"LEX v1.6.0 is the initial model which has completed the initial accuracy tests and been validated for the hybrid model training. However, now it is still an idealized model which does not contain the microphysics and radiation scheme. This LEX version has already included a surface flux scheme following Neale et al. (2010), though it was not tested in this study. The implementation of P3 microphysics (Milbrandt et al., 2021) is ongoing and will be tested in cloudy boundary layer cases. We will also add the Rapid Radiative Transfer Model (RRTMG) (Iacono et al., 2008) to LEX in the future."

**Comment 1.10**

There is no multi-GPU benchmarks in the paper. I believe JAX with XLA can scale across many GPU's, does LAX also scale beyond a single device? Please elaborate in the performance section.

**Response:**

The reviewer is correct that the present study demonstrates the model's performance on a single GPU and does not include multi-GPU or large-scale parallel simulations. Our original intention was to highlight that the model is designed with inherent support for parallelism thanks to its implementation using the JAX framework, which provides automatic parallelization capabilities with the Single-Program Multi-Data (SPMD) codes (referenced from: https://docs.jax.dev/en/latest/sharded-computation.html), as well as mpi4jax (Hafner et al., 2021). The SPMD parallelism can no doubt be effectively applied at the batch level, and we have uploaded the codes to realize batch-level parallelism for model training. However, SPMD parallelism across the spatial domain is not plausible due to the usage of ghost points when we simulate subdomains, and mpi4jax is the appropriate tool for domain decomposition and parallel computing. Our current priority is to develop other physics modules (microphysics, radication, etc.). To further explain these in the manuscript, we have added related elaborations in the conclusion part.

Manuscript text (Lines 411–416):

"Moreover, the LEX model is designed with inherent support for parallelism thanks to its implementation using the JAX framework, which provides automatic parallelization capabilities with the Single-Program Multi-Data (SPMD) codes (Bradbury et al., 2018), as well as mpi4jax (Häfner et al., 2021). However, SPMD parallelism across the spatial domain is not plausible due to the usage of ghost points when we simulate subdomains, and mpi4jax is the appropriate tool for domain decomposition and parallel computing. Now the parallelism of LEX v1.6.0 can only be performed at the batch level. Related codes are provided on Github."

There is no mentioning of hyperparameter choices or tuning thereof in the SGS training section. It could be nice to add a small exploration of this.

**Response:**

The updated AE model is trained for 88 epochs in total. The training process is manually tuned based on the observed loss. The look-ahead step starts at 3 and progressively increases to 6 and 12, while the learning rate is decayed from 1e-3 to 1e-6. This information has also been added in section 2.2.2 and the loss plot has been added to supplement as Figure S1 for readers' reference.

Manuscript text (Lines 239–241):

Figure S 1: Strong scaling performance of CM1 on the AMD Ryzen Threadripper 3990X.

"In this paper, the AE model is trained for 88 epochs, with the training process being manually fine-tuned based on the observed loss trajectory. The learning rate is decayed from an initial value of 1e-3 to a final value of 1e-6, which can be seen in Figure S1."

**2 Response to Reviewer 2**

**Major comments**

**Comment 2.1**

There are statements throughout the manuscript which are scientifically imprecise or lack supporting evidence. For example:

- lines 5 & 6" "Thus, developing SGS turbulence models for the gray zone requires new LES models, which ... enable new approaches to develop SGS models". I do not understand what the authors mean by this? To my understanding, an LES model already contains a SGS model. Therefore, how can a new LES model enable the development of an SGS model?
- Line 8: "The new LES model is capable of adequate parallelism ...". How is this claim supported? To my understanding, the LEX model is only run on a single GPU, and parallel simulations are not discussed at all.
- 21 & 22: "The capability of LESs to simulate small-scale turbulence motion...". This sentence feels scientifically imprecise. LES is supposed to resolve large-scale motions while only modeling the effect of small-scale turbulence on aforementioned large eddies.
- Line 40: "GPU codes are known to run much faster than conventional Fortran or C codes on CPUs". In my opinion, this statement is too generic and scientifically imprecise. While GPUs leverage massive thread-level parallelism, achieving actual code speed up is highly dependent on specific applications.
- Lines 68 & 69: "Existing studies have also shown that JAX-GPU codes enable ... less computational costs when the problem sizes become quite large". What do the authors mean by this? Surely, the computational cost can not decrease with increasing problem size? What does "quite large" mean?
- Line 298: "... float64 convolutions are not supported by XLA now,...". In my opinion, this is not true. float64 support is backened (i.e. hardware) specific. The NVIDIA A6000 GPU does in fact not natively support float64. However, NVIDIA A100 or H100 GPUs provide float64 support on the hardware side. Please correct this statement.

**Response:**

Thanks for these questions, our responses and revisions of the article are listed below:

**• Line 5 & 6:**

The reviewer is correct that a standard LES framework inherently contains an SGS model. We apologize for the lack of clarity in our original phrasing, which caused confusion. Our intended meaning was not that a new LES model would contain or directly generate another SGS model. Rather, we aimed to convey that developing new SGS models for the **gray zone** first requires a novel type of computationally efficient LES framework. This platform must be capable of rapid, large-domain simulations to facilitate the testing of the new SGS model, and also, in this paper, enable a data-driven SGS model trained by the hybrid approach, which needs it to allow automatic differentiation. In other words, a new, high-performance modeling framework is a prerequisite that will enable the subsequent development of new SGS

models for gray-zone simulations. Following the reviewer's suggestion, we have revised the manuscript to clarify this point. The amended text now reads:

**Manuscript text (Lines 5–6):**

"Thus, a novel LES framework is required to enable the development of new SGS approaches for the gray zone."

**• Line 8:**

It is correct that the present study demonstrates the model's performance on a single GPU and does not include multi-GPU or large-scale parallel simulations. Our intention was to highlight that the model is designed with inherent support for parallelism thanks to its implementation using the JAX framework, which provides automatic parallelization capabilities with the Single-Program Multi-Data (SPMD) codes (referenced from: https://docs.jax. dev/en/latest/sharded-computation.html) as well as mpi4jax (Hafner et al., 2021). However, we agree that the original phrasing ("is capable of adequate parallelism") could be misinterpreted as a claim that we have already tested and scaled it across multiple devices, which was not the focus of this paper before, because we cared more about the novel parameterization model. Also, though the parallelism can be effectively applied at the batch level, and the related codes have been provided on Github, now SPMD parallelism across the spatial domain is not plausible due to the usage of ghost points when we simulate subdomains, and mpi4jax is the appropriate tool for domain decomposition and parallel computing. Our current priority is to develop other physics modules (microphysics, radication, etc.). To fix this misunderstanding and further give explanations, we have deleted this sentence in the abstract and added related elaborations in the conclusion part.

**Manuscript text (Lines 411–416):**

"Moreover, the LEX model is designed with inherent support for parallelism thanks to its implementation using the JAX framework, which provides automatic parallelization capabilities with the Single-Program Multi-Data (SPMD) codes (Bradbury et al., 2018), as well as mpi4jax (Häfner et al., 2021). However, SPMD parallelism across the spatial domain is not plausible due to the usage of ghost points when we simulate subdomains, and mpi4jax is the appropriate tool for domain decomposition and parallel computing. Now the parallelism of LEX v1.6.0 can only be performed at the batch level. Related codes are provided on Github."

**• Line 21 & 22:**

The reviewer is correct to point out this fundamental principle of LES. Our original phrasing blurred the critical distinction between resolving large-scale motions and modeling the effects of sub-filter-scale (SGS) turbulence. We have revised the sentence to precisely reflect the methodology of LES, acknowledging that it resolves the large-scale eddies and parameterizes the influence of the smaller, unresolved scales. The modified text now reads:

**Manuscript text (Lines 19–22):**

"... The capability of LES to resolve large, energy-containing turbulent eddies and model effects of SGS processes on these resolved scales, as well as their interactions with other processes such as clouds and radiation, makes it a unique and valuable tool in atmospheric science."

**• Line 40:**

We agree that the performance advantage of GPUs is not universal and is indeed highly contingent upon the application's suitability for massive parallelism and the quality of the implementation. Our original statement was overly broad. Our intention was to highlight the significant speedup with GPU acceleration for atmospheric models. The cited references (Demeshko et al., 2013; van Heerwaarden et al., 2017, etc.) specifically demonstrate significant speedups for atmospheric modeling codes when ported to GPUs. There have also been much existed work showing the outstanding performance of GPU codes, such as Sun et al, 2018; Sun et al, 2023, etc.

To reflect the reviewer's valid point and provide a more precise and scientifically robust statement, we have revised the text to include the necessary context and conditioning. The modified sentence now reads:

**Manuscript text (Lines 39–41):**

"For computationally intensive, highly parallelizable applications like atmospheric models, GPU-accelerated codes have been demonstrated to run much faster than conventional CPU-based implementations in Fortran or C (Demeshko et al., 2013; Price et al., 2014; Schalkwijk et al., 2015; van Heerwaarden et al., 2017; Sun et al., 2018, 2023). ..."

**• Line 68 & 69:**

The reviewer is correct that computational cost cannot be decreased as the problem size becomes larger, which is an imprecise expression here. The comparisons for computational costs are contingent upon specific research problems. Upon careful consideration, we deleted this sentence from the manuscript.

**• Line 298:**

Yes, float64 is supported by some hardware side. However, it is XLA (Accelerated Linear Algebra) that doesn't support 64-bit convolutions, which is software or compiler side, and does not conflict with the hardware side. This statement comes from JAX's official documents, which can be referenced from https://docs.jax.dev/en/latest/notebooks/Common\_Gotchas\_in\_JAX.html:

**Comment 2.2**

What is the motivation to choose the conventional Smagorinsky model as a baseline for comparison? It is well known that the dynamic Smagorinsky model outperforms the classical model in many scenarios. This would be a much stronger baseline for benchmarking the DL-based SGS model.

**Response:**

We thank the reviewer for this insightful suggestion. We fully agree that the dynamic Smagorinsky model (DSM) generally provides more accurate results than the standard model and represents a more advanced baseline.

Our choice of the conventional Smagorinsky model was primarily motivated by the specific scope and focus of this initial study. The primary objectives here are: (1) to introduce and validate the new LEX model itself, and (2) to conduct the pretraining with the hybrid modeling approach. The conventional Smagorinsky model, with its simplicity, numerical stability, and low computational cost, serves as a clear and stable benchmark for this purpose. It effectively illustrates the performance level of a widely recognized, first-generation SGS parameterization, thereby clearly highlighting the limitations we aim to address with our new method.

Implementing and thoroughly testing the DSM will be valuable, but it would introduce additional complexity, which is beyond the core focus of this paper. We envision the DSM as an excellent benchmark for the future and thank the reviewer for this, which we may incorporate into our future work.

**Comment 2.3**

I have the following comments and questions regarding model training:

- (a) What is the time step size of the coarse-grained simulation? The time step size of the high-resolution simulation is 5s. While the spatial coarse-graining factors are explicitly mentioned, the authors do not mention whether temporal coarse-graining is also applied.
- (b) What is the rationale for choosing a 6x CG in the horizontal direction and a 3x CG in vertical direction?
- (c) Please specify the loss functional explicitly. Specifically, the mean-squared error of which quantities is used?
- (d) Please provide more information regarding hyperparameters of the model training. How many optimization steps are used during training? What is the final training loss level? What is the stopping criterion? What is the learning rate? Is there a learning rate scheduler?
- (e) The authors mention, that the training for the dry case can "achieve asymptotic convergence" while it "shows oscillatory convergence behavior" for the moist case. Please add loss plots for both scenarios to the manuscript (e.g., to the appendix).
- (f) Are the authors using custom implementations to propagate the AD gradients through the BiCGSTAB solve?

**Response:**

Thanks for the comments. Here are our reply:

• (a) Temporal coarse-graining is not applied in the original submitted version of this work. The time step for low-resolution training is still 5 seconds before. However, in the latest version, we have applied temporal coarse-graining with a 15-second time step in the training and testing processes, achieving faster simulation speed as well as good numerical stability. We have added this temporal information in the revised manuscript.

Manuscript text (Lines 186–187):

"Temporal coarse-graining is employed in the training and testing processes, with a 15-second time step for the numerical simulation."

- (b) This is because for the real atmosphere, the horizontal scale is typically much larger than the vertical scale, therefore the vertical grid spacing is typically smaller. The chosen coarse-graining factors (6x horizontal, 3x vertical) are intended to roughly reflect this common practice.
- (c) In our updated code version, we added a loss term to penalize negative values of water vapor generated by the DL model, and a Laplacian loss term to reduce background noise.

Scaling parameters are also applied to the outputs of the DL model and the loss terms for further improvement. Now the loss function contains three parts: the L2 loss, the Laplacian loss, and the penal-neg loss. For the L2 loss, it is calculated with the density potential temperature perturbation  $(\theta')$ , pressure perturbation  $(\pi')$ , mixing ratio of water vapor perturbation  $(q_v)$ , horizontal and vertical velocity (u, v, w). For the Laplacian loss, it is calculated with  $\theta'$  and  $q_v$ . And for the penalize-neg loss, it is only calculated with  $q_v$ . Related information has been provided in section 2.2.2.

**Manuscript text (Lines 205–238):**

"At each step, the L2 loss, Laplacian loss, as well as an extra loss term which is used to penalized unreasonable model outputs are employed and accumulated to be the total loss of the current training step, with which we use the Adam (Kingma and Ba, 2017) optimizer to adjust the DL model parameters. In this study N = 12. To mitigate the influence of the potential rounding errors, doubleprecision (float64) is employed throughout the training process.

The  $L_2$  loss is written as:

$$\mathcal{L}_{l_2} = \frac{1}{2} \| \mathbf{M}^k \mathbf{x}_{t_0} - \mathbf{x}_{t_k} \|_2^2,$$

 $\mathcal{L}_{l_2} = \frac{1}{2} \| \bar{M}^k x_{t_0} - x_{t_k} \|_2^2,$  where x is the tensor of the general physical state consisting of density potential temperature perturbation ( $\theta'$ ), mixing ratio of water vapor perturbation (qv'), horizontal and vertical velocity (u, v, w). M represents the LEX model (dynamical core and the SGS model, if any),  $x_{t_0}$  is the initial state of x, and  $x_{t_k}$  is the truth state of x at the  $k^{th}$  look-ahead step during training.

When training, it is found that the DL model can not distinguish the physical meaning of the input variables and will generate unreasonable outputs in the very beginning, such as negative values for water vapor, and meaningless background noise, which are against the laws of physics and will strongly impact the integrated results of the numerical simulation in the next time step during the training loop.

Thus, firstly, to penalize such unreasonalbe negative values that may be caused by the DL model, an extra loss term is added, which is:

$$\mathcal{L}_{\textit{penal-neg}} = \begin{cases} -\textit{qv}_{i,j,k}, & \textit{if } \textit{qv}_{i,j,k}

Figure S 1: Strong scaling performance of CM1 on the AMD Ryzen Threadripper 3990X.

"In this paper, the AE model is trained for 88 epochs, with the training process being manually fine-tuned based on the observed loss trajectory. The learning rate is decayed from an initial value of 1e-3 to a final value of 1e-6, which can be seen in Figure S1."

- (e) The loss plot has been shown and been added to the supplement.
- (f) Not really. We did not use a custom implementation. The gradients through the BiCGSTAB solver are handled entirely by the high-level interface provided by the JAX framework, the jax.scipy.sparse.linalg.bicgstab function, which can be referenced from: https://docs.jax.dev/en/latest/\_autosummary/jax.scipy.sparse.linalg.bicgstab.html. Leveraging this built-in functionality ensures both the correctness of the gradient computation and the numerical stability of the adjoint method, which might be challenging to achieve with a custom implementation.

**jax.scipy.sparse.linalg.bicgstab**

```
jax.scipy.sparse.linalg.bicgstab(A, b, x0=None, *, tol=1e-05, atol=0.0,
maxiter=None, M=None) [source]
```

Use Bi-Conjugate Gradient Stable iteration to solve Ax = b.

The numerics of JAX's bicgstab should exact match SciPy's bicgstab (up to numerical precision), but note that the interface is slightly different: you need to supply the linear operator A as a function instead of a sparse matrix or LinearOperator.

As with cg, derivatives of bicgstab are implemented via implicit differentiation with another bicgstab solve, rather than by differentiating *through* the solver. They will be accurate only if both solves converge.

**Comment 2.4**

I have the following comments and questions regarding the chosen parameterization and DL model:

- The standard WENO3- and WENO5-JS schemes are known to be overly dissipative. What is the motivation to choose this parameterization?
- The DL-SGS model output is applied to  $\theta$ , u, v, w and qv. Does the same hold true for the Smagorinsky model? Is the mixing ratio of water vaport qv a transported quantity or is it post-processed?

**Response:**

Thanks a lot for the comments. Here are our reply:

Our primary motivation was numerical robustness and stability, especially for long-time integration of turbulent flows in the atmospheric boundary layer, which can develop sharp gradients and discontinuities. The WENO-JS scheme provides a proven and computationally efficient mechanism to eliminate spurious numerical oscillations. We deemed the guaranteed stability of WENO-JS to be a critical advantage. Thanks for this question and we have added this explanation to the manuscript in section 2.1.2.

Manuscript text (Lines 134–135):

" ... The WENO scheme provides a proven and computationally efficient mechanism to eliminate spurious numerical oscillations. ...

• Yes, it's the same for the Smagorinsky model, although the viscosity for momentum differs from the diffussivity for scalars. qv is a transported quantity.

**Comment 2.5**

Validation of the LEX solver with CM1 results are purely qualitative. Please add quantitative comparisons if possible. The authors mention that "results of the LEX are identical with those of CM1". This is an overstatement in my opinion, as Fig. 2 shows visible discrepancies between the two simulations. For example, the lower parts of the thermal are clearly different at later times and the structure of the rotors show differences. The authors should tone down this claim or provide quantitative evidence for it.

**Response:**

Figure 3: The correlation coefficient (R), mean squared error (MSE), and multi-scale structural similarity (MS-SSIM) for CM1 and LEX simulation results of density potential temperature (a, c, e), and water vapor mixing ratio (b, d, f).

Thanks a lot for the comment. We have added Figure 3 in the revised manuscript, which presents the correlation coefficient (R), mean squared error (MSE), and multi-scale structural similarity (MS-SSIM) between CM1 and LEX. Results show that the simulations of LEX are highly identical with CM1 before the warm bubble reaches the top boundary, with R and MS-SSIM being above 0.99, and MSE keeping being almost zero. After that, as has been mentioned in the manuscript, because LEX calculates pressure based on the pseudo-compressible approximation, subtle differences appear after the thermal reaches the upper boundary of the domain in pressure simulations. Thus, the last

figures look a little bit different, and the evaluation metrics also present an obvious change after the time the bubble reaches the upper boundary. Now the related words have been:

Manuscript text (Lines 247-255):

"Comparing the two pairs in Figure 2, it is evident that the simulation results of the LEX model demonstrate excellent agreement with those of CM1, regardless of the initial potential temperature perturbations, indicating the reliability and accuracy of the LEX model code in JAX. Also, results shown in Figure 3 demonstrate excellent performance metrics of LEX. The correlation coefficient (R) and multi-scale structural similarity (MS-SSIM) maintain high values and mean squared error (MSE) maintains low levels throughout the simulation period for two tested cases. Figure S2 and Figure S3 further confirm the robustness of the LEX model by presenting the simulated results for the mixing ratio of water vapor and pressure perturbations with  $\theta_c = 1$ K and  $\theta_c = 5$ K. However, because LEX calculates pressure based on the pseudo-compressible approximation, subtle differences appear after the thermal reaches the upper boundary of the domain in pressure simulations.

**Comment 2.6**

I would encourage the authors to verify the implementation of the AD gradients with finite-difference analogs. A simple test case can be chosen in which only a single parameter of the DL-based model is optimized with AD and with FD. The error between the two should converge as the step size in the FD approximation approaches zero.

**Response:**

Thank you for this suggestion. However, after careful consideration, we think the chosen JAX framework can already provide mathematical guarantees. Firstly, the AD implementation in this paper used <code>jax.grad</code> to realize. This is not a numerical approximation but is a method for computing analytical gradients based on the chain rule. More information about JAX's AD can be accessed from <code>https://docs.jax.dev/en/latest/automatic-differentiation.html</code>. Also, JAX has been developing for several years till now, and has been widely used and examined by numbers of scientific projects. Its developers have also verified its mathematical correctness strictly. Therefore, using JAX's AD is considered a reliable and standard practice in the machine learning field, and its validity does not typically require re-verification by end-users for each specific application. What's more, though we does not provide the direct AD-FD test, the successfull convergence of the updated AE model can be another form of validation.

**Comment 2.7**

The authors mention the trained DL model can "develop the proper symmetric structure of the thermal". This statement is not true. Figs. 4 and 6 clearly show that the DL model breaks symmetry. The authors themselves acknowledge this fact later in Section 4.2.2.

**Response:**

We agree with the reviewer that in the initial submitted article, the predicted structure of the hybrid model was not perfectly symmetric. We have now updated our DL model with an autoencoder (AE) model (see Figure 1), whose model structure is attached below and also updated in the manuscript. The loss functions (detailed in the response to Comment 2.3 above) have also been revised to improve the hybrid model's performance. The updated AE model runs faster and is validated to keep the bubble's structure almostly strictly symmetric for the moist case (Figure 5 and

Figure S5).

Figure 1: Model Architecture for the three-dimensional autoencoder neural network, where  $a \times b \times c \times d$  means width×length×height×channel. The inputs include the density potential temperature perturbation ( $\theta'$ ), pressure perturbation ( $\pi'$ ), mixing ratio of water vapor perturbation ( $q'_v$ ), horizontal and vertical velocity (u, v, w), and the outputs are SGS corrections for the density potential temperature ( $\theta$ ), mixing ratio of water vapor ( $q_v$ ), horizontal and vertical velocity (u, v, w).

Figure 5: Snapshots of simulated potential temperature perturbations ( $\theta'$ ) at t = 0, 5, 10, 15, and 20 min, with  $\theta_c = 2.6K$  (the first to the third columns), and  $\theta_c = 5.0K$  (the fourth to the sixth columns), where the first and fourth columns are the 'Coarsened Truth' simulations with a coarse resolution of  $600 \times 600 \times 300m$ , the second and fifth columns are the 'LowRes-Smag' simulations with the Smagorinsky scheme to deal with the SGS turbulence, and the third and sixth columns are 'LowRes-DL' simulations with the trained AE model to serve as the turbulence parameterization scheme.

Figure S 5: Snapshots of simulated water vapor mixing ratio  $(q_v)$  at t = 0, 5, 10, 15, and 20 min for moist cases, with  $\theta_c = 2.6K$  (the first to the third columns), and  $\theta_c = 5.0K$  (the fourth to the sixth columns), where the first and fourth columns are the 'Coarsened Truth' simulations with a coarse resolution of  $600 \times 600 \times 300m$ , the second and fifth columns are the 'LowRes-Smag' simulations with the Smagorinsky scheme to deal with the SGS turbulence, and the third and sixth columns are 'LowRes-DL' simulations with the trained AE model to serve as the turbulence parameterization scheme.

**Comment 2.8**

It is mentioned that the mixing ratio of water vapor has to be clipped after application of the DL model (Section 2.2.2). I am interested how often this occurs for the trained model over the course of a simulation.

**Response:**

Before for each correcting step the clipping procedure should be done, but with the updated AE model and loss function (see details in the response to Comment 2.3 above and Figure 5 and Figure S5 in the manuscript), now there's no need to clip water vapor during the training process as well as the validation simulations any more.

**Comment 2.9**

I agree with the authors that the DL-based SGS model outperforms the conventional Smagorinsky model for the thermal test case. To my understanding, the DL model is applied after a full integration step while the Smagorinsky model is applied per stage (i.e., thrice per integration step). Can the authors elaborate on this? It would be very interesting to visualize the output of the DL model to try to understand its improved SGS modeling capabilities. Have the authors done such analyses? Is the model output interpretable? What conclusions can be drawn from it?

**Response:**

This is not exact for the Smagorinsky model. We only calculate the Smagorinsky tendencies once, and then they are kept as constant and applied in sub-steps of the SSPRK3 integration.

We sincerely thank the reviewer for this constructive suggestion. We have visualized the SGS corrections generated by the classic Smagorinsky and the DL model. Updated comparisons and analyses have been added to a new Section '4.3 Comparisons and Potential Physical Insights'.

Figure 7: The simulated potential temperature perturbation ( $\theta'$ ) at t = 5, 10, and 15 min, with  $\theta_c = 2.6K$ . The first and the second columns are the forecasts of conventioanl Smagorinsky and DL-based SGS model. The third and the fourth columns are the differences between parameterized and non-parameterized simulation results of the conventional Smagorinsky and the DL-based SGS model. The fifth column is the SGS corrections generated by the DL model.

Figure 8: The simulated vertical velocity (w) at t = 5, 10, and 15 min, with  $\theta_c = 2.6K$ . The first and the second columns are the forecasts of conventioanl Smagorinsky and DL-based SGS model. The third and the fourth columns are the differences between parameterized and non-parameterized simulation results of the conventional Smagorinsky and the DL-based SGS model. The fifth column is the SGS corrections generated by the DL model.

**Manuscript text (Lines 304–337):**

"In this section, the differences between the parameterized and non-parameterized simulations of the conventional Smagorinsky scheme and the AE model are compared, and the SGS corrections generated by the AE model are analyzed, aiming to find the potential reasons that the conventional Smagorinsky scheme fails to develop the correct rotor structure, and the difference that the hybrid model has brought. Through this way, we hope to give some physical insights from the DL-based SGS model and make some contributions to the development of the interpretable DL.

The warm bubble case is set with an initial temperature perturbation, which causes an upward buoyancy and thus gives the bubble a vertical acceleration. When the bubble rises, the cold air on each side needs to descend for compensation, which will then cause vertical velocity gradients and further form strong velocity shear layers at the bubble boundaries. In the shear layers, according to the vorticity equation, vorticity will thus be generated due to the spatial gradients of potential temperature, which is  $\nabla \theta$ . But when the resolution becomes coarse, small-scale processes and some key physics information, such as temperature gradients, cannot be appropriately resolved, and this causes the LowRes simulations to be unable to generate the rotor structure.

Figure 7 and Figure 8 present the forecasted potential temperature perturbation and vertical velocity from the conventional Smagorinsky scheme and the AE model, respectively. The forecast differences induced by each scheme, and the corresponding SGS corrections generated by the DL model are also shown. Results for the additional physical quantities  $(u, v, and q_v)$  are provided in the supplement (Figure S6, S7, and S8).

As evidenced by Figure 7 and Figure 8, the Smagorinsky scheme and the AE model exhibit obviously different impact on the development of the warm bubble at the very beginning. The Smagorinsky mainly imposes a cooling effect to the warm bubble, and weakens its upward motion. But the AE model sustains warming and the upward motion in regions that are to further develop the rotor structure. This significant difference is key to the later development of the warm bubble.

As the conventional Smagorinsky is a diffusion model, it naturally diffuse warm temperature anomaly to surrounding regions. However, the diffusivity of the Smagorinsky model is larger when the flow deformation is strong, so the diffusion mainly happens below the warm bubbles' top, where wind shears exist. It produced a cooling effect near the top of the rising thermal and warming effect below. This can explain Figure 4, where the classic Smagorinsky helps correct the warm bubble's rising speed compared to the LowRes results, as the Smagorinsky scheme greatly lowers down the temperature at the top. Furthermore, it aligns with Figure 6 and Figure 8, where the Smag forecast has the same energy and vertical velocity peak with the hybrid model, but it presents smaller values, even smaller energy than the LowRes simulation in Figure 6.

However, the classic Smagorinsky tends to produce overly diffusive corrections, which limits it to resolve fine-scale structures and maintain the necessary energy for the warm bubble to develop the rotor structure. The corrections generated by the AE model are much more detailed and accurate. As is illustrated in Figure 7, SGS corrections of the AE model always help maintain the strength of the potential temperature at the critical part of the warm bubble in a very fine way, such as the rising top at the key beginning, and the rotors on the sides after they have been maturely developed. This makes the hybrid model keep the energy for rising and developing the rotors. These detailed structures are probably essential to enable the model to model the small-scale physics information which is unresolvable by the coarse grid. Similarly, Figure 8 shows that the AE model's corrections exhibit detailed structures and help keep the upward motion.

**Comment 2.10**

I have the following comments and questions regarding the computing time comparison:

- The performance comparison in Section 5 is somewhat misleading. The authors claim that they achieve a 92:1 speed up when comparing the LEX code run on an A6000 GPU with the CM1 code run on a single CPU core. I think the authors are aware that such a comparison is not meaningful at all. Can the authors comment on this?
- In Section 5.2, the wall-clock time of the DL-based SGS model is compared with the Smagorinsky model. Given the short simulation time, the wall-clock time measurements are strongly influenced by the duration of the just-in-time compilation. I would encourage the authors to simply evaluate the Smagorinsky model and the DL-based SGS model on their own to provide more meaningful WCT measurements or to exclude the duration of the jit-compilation from the performance measurements.

**Response:**

Thanks a lot for the suggestions. Here are our reply:

We thank the reviewer for this important comment, which rightly challenges the meaningfulness of our original performance comparison. We admit that it is rare to use only one CPU core to run CM1. In Table 1, we also provided the benchmark simulation result with 64 cores. Our initial goal was to provide a conceptual baseline by adding the comparison for one GPU and one CPU, and to illustrate the equivalent number of CPU cores compared to one GPU when

applying different time steps, like 12 seconds for LEX and 2 seconds for CM1, as we don't have so many computational resources to conduct a real test. However, we recognize that comparing one GPU to a single core of a CPU is not a fair representation of this concept. Also, by conducting the strong scaling test (the newly added Figure 9), we find that 64 cores will not bring a linear speedup over one core, which means 92 is also not the right number. Furthermore, the statement in Line 337, which claims 'running LEX on one GPU is as fast as running CM1 on 600 CPU cores', is also not real, because the speedup is found to achieve the maximum with 64 cores. As a result, we have deleted these two comparisons and the revised paper by comparing 64 CPU cores and one GPU, which is around nine times faster for LEX with one GPU compared to CM1 with 64 CPU cores, using the time step of 12 seconds and two seconds, respectively. Our revised manuscript regarding the computational cost comparison for LEX and CM1 has been shown below.

Figure 9: Strong scaling performance of CM1 on the AMD Ryzen Threadripper 3990X.

**Manuscript text (Lines 340-356):**

"The computational costs are compared in this section. As mentioned in Section 1, LEX has better numerical stability and is expected to show faster computing speed with JAX acceleration techniques. Using the conventional CM1 model as the benchmark model, Table 1 shows that employing the same time step of two seconds to run a 20-minute simulation, the total computing time for LEX is 789 s using 64 cores, while the LEX run takes 548 s on one GPU. Furthermore, at the resolution of  $100 \times 100 \times 100$  m, the longest time step for CM1 to maintain numerical stability is two seconds, but for LEX, it can be up to twelve seconds, thanks to its acoustic-wave-filtering equations and the strong stability integration scheme SSPRK3. As a result, LEX's running time can be further reduced by a factor of 1/6. Meanwhile, according to the strong scaling test shown in Figure 9, the speed-up factor for the 20-minute simulation of CM1 reaches the maximum with 64 processors. That means in this 20-miniute simulation for the warm bubble case, compared to the optimal speed-up performance of CM1 with 64 CPU cores, LEX on a single GPU is around nine times faster.

Because the 20-minute simulation is a relatively short integration period, leading to the LEX setup and just-in-time compilation time accounting for a significant fraction of the total running time. However, if we run the LEX for a substantially longer time, the compilation and setup time probably can be ignored. This demonstrate the great application potential of LEX to run for long simulations.

The effectiveness of GPU acceleration is also shown in Table 2. Calculating with the same resolution and a 15-second time step for a 20-minute integration time, LEX with the Smagorinsky scheme runs around 21 times faster on the GPU than on the CPU, excluding the just-in-time compilation time."

• Thanks a lot for this suggestion. We have now updated Table 2 with the IO/Setup time, compilation time, and computing time listed separately. As the newly used AE model allows a 15-second time step, now in this revised version, all the computing time is tested with this time step. Related analyses have also been updated in section 5.2. Moreover, as it is found that the newly trained model with the AE model and the updated loss function can conduct forecasts with the single precision of float32, comparisons and analyses for the mixed precision mode has been deleted. But we still includes the double precision result for the hybrid model as a note for the reader that float64 convolutions are not supported by XLA now.

**Table 2.** Computational speed comparison of DL-based SGS model and conventional Smagorinsky Scheme, with the resolution being  $600 \times 600 \times 300 \ m$ , and the 15-second time step for a 20-minute simulation test for each.

| Model         | Hardware | Parameterization | IO/Setup | Compilation | Execution | Model Inference |
|---------------|----------|------------------|----------|-------------|-----------|-----------------|
|               |          | Scheme           | (s)      | (s)         | Time (s)  | Time (s)        |
| LEX           | GPU      | N/A              | ~5       | ~31         | 0.89      | N/A             |
| LEX           | CPU      | Smagorinsky      | ~5       | $\sim$ 28   | 43.28     | N/A             |
| LEX           | GPU      | Smagorinsky      | ~5       | ~28         | 1.91      | 1.02            |
| LEX+DL (fp32) | GPU      | DL               | $\sim 5$ | $\sim 60$   | 1.48      | 0.59            |
| LEX+DL (fp64) | GPU      | DL               | $\sim 5$ | $\sim 60$   | 6.18      | N/A             |

**Manuscript text (Lines 357–375):**

"LEX can be trained with a DL-based SGS model and succeed in numerical predictions in the gray zone, but whether such physics-DL hybrid models can be applied in real weather forecasts also relies on their computational costs. The parameterizations for SGS processes are only one part of the entire numerical weather predictions, thus, they are expected to run at a fast speed. Since the DL model is trained with the double-precision float64, its computing time is first evaluated with the same precision to run the hybrid model. Table 2 shows that when running with float64, the LEX-DL model with a 15-second time step takes around three times of the computing time of the LEX-Smag model using float32 with a same time step after compilation, and meanwhile its compilation time is two times slower, which is not satisfying performance. One reason for this is float64 needs more computational resources than float32, and the other is float64 convolutions are not supported by XLA now, which further increases its computational costs.

However, though the double precision is necessary for the training of the LEX-DL model, a single-precision of float32 is found to be applicable for the evaluations, as the model parameters have already been sufficiently trained and the DL model will not cause some tiny noise towards the stable thermal structure. Thus, the computing efficiency of the DL-based SGS model is further enhanced. As is shown in Table 2, using the same time step of 15 seconds, the LEX-DL model with a single precision can achieve 76% computing time reduction than that with the double precision, which only needs 1.48s to complete the integration task after the compilation.

A further comparison is also conducted and it is found that though the compilation time is two times slower, the fastest speed the hybrid model can achieve now after compilation is faster than that of the LEX-Smag model with the single precision, which means the DL model can enable a lower computational expense for prolonged forecasts."

**Comment 2.11**

The authors should consider citing JAX-Fluids [1, 2] and [3]. JAX-Fluids is a JAX-based fully-differentiable CFD solver for compressible single- and two-phase flows, which is closely connected with the present research. Specifically, JAX-Fluids implements functionality for LES and has been used for end-to-end training of implicit LES models [3].

**Response:**

Thanks a lot, the references have been added.

Manuscript text (Lines 68–70):

"Existing work includes JAX-Fluids, a Python-based end-to-end differentiable CFD framework which is designed with JAX for compressible single and two-phase flows (Bezgin et al., 2023, 2025a), and enables end-to-end training of DL-based implicit LES models (Bezgin et al., 2025b)."

**Minor comments**

**Comment 2.12**

What is the reason for v1.4 in the title of the manuscript? Maybe I have missed it, but it is not mentioned in the remainder of the paper. Is the present work building upon a previous release of the LEX solver?

**Response:**

LEX v1.4 was the first released version for public use. Before it was published, which was also during the DL training process, we had fixed some initial errors found during this period. When writing this paper, we called it v1 at first. However, the version in GitHub was already v1.4 at that time, and the editor required us to use the same version number for the paper. This is why it is v1.4 in the title. Moreover, as we have updated our work, the code version has also been updated to v1.6.0 now. In our next paper, when we complete some other sections like microphysics, radiation, etc, maybe we will use LEX v2.x as the title.

**Comment 2.13**

In section 2.1.1, some variables are not defined, including  $\epsilon$ , cp, cv, w, g, ps, R. While I assume that many of these quantities are well known (presumably, cp is the heat capacity at constant pressure), it would improve clarity to specify their definition once.

**Response:**

Thanks a lot for this comment. It is our ignorance. To enable more readers, especially those who may lack some professional background, to get a better understanding, we have now defined all the listed variables ( $\epsilon$ , cp, cv, w, g, ps, R) upon their first appearance in Section 2.1.1 to ensure the paper is accessible to a broader audience.

Manuscript text (Lines 95):

"... where  $\varepsilon = R_d/R_v$ ,  $R_d$  and  $R_v$  are gas constants for dry air and water vapor, respectively. ..." Manuscript text (Lines 100): "...  $c_p$  is the specific heat of air at constant pressure, ..."

Manuscript text (Lines 109):

"... w is the vertical velocity, ..."

Manuscript text (Lines 113):

"... and g is the gravitational acceleration. ..."

Manuscript text (Lines 117):

"... where R is the gas constant for dry air,  $c_v$  is the specific heat of air at constant volume, and  $p_s$  is the pressure at the referenced level. ..."

**Comment 2.14**

Please define the correlation coefficient R and the kinetic energy KE in Section 4.2.

**Response:**

Thanks a lot for this reminder. Definitions of R and KE have been added.

Manuscript text (Lines 288–295):

"The quantitative assessments of the DL model's forecast performance are also conducted with the correlation coefficient (R) and the kinetic energy (KE) profile, which are defined as:

$$R = \frac{\sum_{i} \sum_{j} \sum_{k} (X_{ijk} - \overline{X}) (Y_{ijk} - \overline{Y})}{\sqrt{(\sum_{i} \sum_{j} \sum_{k} (X_{ijk} - \overline{X})^{2})(\sum_{i} \sum_{j} \sum_{k} (Y_{ijk} - \overline{Y})^{2})}},$$

$$KE = \frac{1}{2} \left\langle u_i' u_i' \right\rangle_t,$$

where  $\dot{X}$  represents the simulated results, Y represents the truth states, and the overline denotes the spatial average over all grid points for different variables.  $\langle \cdot \rangle_t$  represents the time average, and  $u'_{\cdot}u'_{\cdot}$  follows the Einstein summation convention, which equals  $u'^2 + v'^2 + w'^2$ . "

**Comment 2.15**

**Technical Correction 1:**

Please proofread and type-check the manuscript carefully. A couple of typos:

- (a) In the gray zone, turbulence and convection ... in line 30.
- (b) the acoustic-wave-filtered equations ... are adopted in line 84.
- (c) for validation simulations. in lines 178 & 179.
- (d) I think the abbreviation LESs is not commonly used.

**Response:**

Thank you so much. All the sentences have been corrected accordingly.

Manuscript text (Lines 29):

" ... In the gray zone, turbulence and convections can only be partially resolved ...

Manuscript text (Lines 85):

" To develop LEX, the acoustic-wave-filtered equations for compressible stratified flow developed by Durran (2008) are adopted as the governing equations, ... Manuscript text (Lines 188–189):

" To validate the trained model, two additional cases with  $\theta_c=2.6\,\rm K$  and 5.0 K are chosen to generate initial conditions for validation simulations. ...

All the 'LESs' have been revised to LES.